# Improved understanding of anthropogenic and biogenic carbonyl sulfide (COS) fluxes in Western Europe from long-term continuous mixing ratios measurements

Antoine Berchet[1, *], Isabelle Pison[1, *], Camille Huselstein[1], Clément Narbaud[1], Marine Remaud[1, 2], Sauveur Belviso[1], Camille Abadie[1, 3], and Fabienne Maignan[1]

[1]Laboratoire des Sciences du Climat et de l'Environnement, CEA-CNRS-UVSQ, Gif-sur-Yvette, France
[2]now at: Faculty of Science, A-LIFE, Vrije Universiteit Amsterdam, 1081 HV Amsterdam, the Netherlands
[3]now at: Discipline of Botany, School of Natural Sciences, Trinity College Dublin, Dublin, Ireland
[*]Correspondance: antoine.berchet@lsce.ipsl.fr, isabelle.pison@lsce.ipsl.fr

**Abstract.** Lack of knowledge still remains on many processes leading to COS atmospheric fluxes, either natural such as the oceanic sources or the vegetation and soil uptakes, or anthropogenic, with emissions from industrial activities and power generation. Moreover, COS atmospheric mixing ratio data are still too sparse to evaluate the estimations of these sources and sinks at the regional scale; in this context, regional estimates are very challenging. This study assesses the anthropogenic emissions and biogenic COS uptakes at the regional scale, in the footprint of a measurement site in Western Europe, at a seasonal to diurnal time resolution over half a decade. The continuous time series of COS mixing ratios obtained at the monitoring site of Gif-sur-Yvette (in the Paris area) from August 2014 to December 2019 are compared to simulations with the Lagrangian model FLEXPART, transporting oceanic sources, biogenic land fluxes from the land-surface models ORCHIDEE and SiB4 and anthropogenic emissions by two different inventories. At GIF, the seasonal variations of COS mixing ratios are dominated by the contributions of the background and ocean; the weekly to daily variations are driven by the biogenic land contribution; anthropogenic emissions may dominate for short episodes of high concentrations. The anthropogenic emission inventory based on reported industrial emissions and the characteristics of coal power plants in Europe is consistent with the observations. The main limitation of this inventory is the flat temporal variability applied to anthropogenic fluxes due to lack of information on industrial and power-generation activity in viscose factories and in coal power plants. As a consequence, there are potential mismatches in the simulated plumes emitted from these hot-spots. We find that the net ecosystem COS uptake simulated by both ORCHIDEE and SiB4 is underestimated in winter at night, which suggests improvements in the parameterization of the nighttime uptake by plants for COS. In spring, SiB4 simulates persistent nighttime uptake by vegetation, contrary to ORCHIDEE, which leads to more realistic simulations with SiB4 than with ORCHIDEE. In summer, both models represent fluxes sufficiently well, with better agreement from ORCHIDEE in terms of magnitudes.

# 1 Introduction

Carbonyl sulfide (COS) is absorbed along $CO_2$ by plants during photosynthesis. But COS, contrary to $CO_2$, is almost not emitted during respiration-like processes (Protoschill-Krebs et al., 1996; Montzka et al., 2007; Sandoval-Soto et al., 2005; Ma et al., 2021; Kuai et al., 2015; Campbell et al., 2017; Kettle et al., 2002; Suntharalingam et al., 2008). This has led to suggestions that COS could be used as a tracer for quantifying and/or reducing uncertainties on $CO_2$ fluxes due to photosynthesis (e.g., Whelan et al., 2018; Launois et al., 2015; Hilton et al., 2017, 2015).

The methodology called atmospheric inversion consists in assimilating mixing ratio data into a statistical framework (called Bayesian because it is based on Bayes' theorem) making use of some prior knowledge of sources and sinks and of a chemistry-transport model (CTM) to link fluxes (either emissions to the atmosphere or uptake) to atmospheric mixing ratios, in order to estimate optimized fluxes. The optimized fluxes are statistically consistent with all the information provided by the prior knowledge of sources and sinks and by the measured atmospheric mixing ratios. The minimum requirements for atmospheric inversion to yield useful insights are therefore *i)* that the CTM resolution, various inputs (such as meteorological fields, flux maps) and physical and chemical parameterizations are relevant to the targeted spatio-temporal scale, so that the link between fluxes and atmospheric mixing ratios does not entail large or poorly characterized uncertainties, but also *ii)* that the uncertainties on the prior knowledge of sources and sinks are well characterized, so that the corrections applied to obtain optimized fluxes are physically meaningful. On top of these general requirements, when using COS mixing ratio data to get information on $CO_2$ uptake during photosynthesis, the atmospheric inversion must be provided with information that make it possible to disentangle the influence of the biogenic sink of COS from the influence of anthropogenic and other biogenic fluxes on the measured COS mixing ratios. This can be done by providing to the CTM fluxes of COS which are well-known, i.e., with small and well-characterized uncertainties. It is therefore a strong limitation to today's potential use of COS to gain insight into $CO_2$ photosynthesis fluxes that COS sources and sinks are not very well known so far, with only a few estimations available for the various categories (Whelan et al., 2018; Remaud et al., 2023a), which can be summarized as (see also Table 1):

- the natural oceanic source, due to both direct COS emissions and indirect emissions via dimethyl sulfide (DMS) and carbon disulfide ($CS_2$) (Mihalopoulos et al., 1992). Note that fresh waters also contribute to COS emissions (Du et al., 2017). In the atmosphere, $CS_2$ is oxidized into COS in about 10 days (Bandy et al., 1981; Khalil and Rasmussen, 1984; Chin and Davis, 1993; Stickel et al., 1993). The total of direct and indirect oceanic emissions are estimated at $265\pm210$ GgS.yr$^{-1}$ by Lennartz et al. (2017) (cited in Whelan et al., 2018) and Lennartz et al. (2021).

- anthropogenic sources of COS, restricted to particular industries, once again in contrast to $CO_2$. These anthropogenic sources are due to industries which emit either COS or $CS_2$. The main source of anthropogenic COS is the oxidation of $CS_2$ emitted by the viscose industry which includes factories producing viscose as their final product (named hereafter viscose-producing industry) or as the by-product of their main process for producing, e.g., sponges, cellophane (Chen, 2015; Water, 2011) (named hereafter viscose processors). Other sources are smaller and emit both COS and $CS_2$: they are in the sector of energy production with coal use in power plants but also the combustion of oil and bio-fuels (Attar, 1978), the pulp mills due to the kraft process (Brownlee et al., 1995; Cheremisinoff and Rosenfeld, 2010), industries

using aluminum oxide electrolysis to produce aluminum (Harnisch et al., 1995), the producers of pigments with carbon black used for tires or the food industry and the widely used white titanium dioxide. For the whole world in the year 2012, the inventory by Zumkehr et al. (2018) gives a total of $400\pm180$ GgS.y$^{-1}$ for all anthropogenic emissions (used in the budget elaborated by Whelan et al., 2018).

– besides the anthropogenic combustion of bio-fuels, biomass burning in open fires either natural (e.g., wild fires due to lightning) or human-caused (e.g., agricultural practices) is also a source of COS, estimated at an average $60\pm37$ GgS.y$^{-1}$ by Stinecipher et al. (2019).

– soils can also be sources of COS under specific conditions such as high temperature and incoming radiation, related to abiotic production processes (Whelan and Rhew, 2015; Whelan et al., 2016, 2018; Kitz et al., 2017, 2020). The anoxic soil contribution has recently been estimated at 96 GgS.yr$^{-1}$ with the ORCHIDEE process-model (Abadie et al., 2022b, see also details on ORCHIDEE in Section 2.2.2).

– the main biogenic sink is due to the uptake of COS by vegetation (Whelan et al., 2018): COS is irreversibly consumed by the carbonic anhydrase enzyme in leaves (DiMario et al., 2016). Soils can also absorb COS due to soil microorganisms that also contain the carbonic anhydrase enzyme (Masaki et al., 2021). The order of magnitude obtained for the uptake by vegetation is -530 GgS.yr$^{-1}$ with the ORCHIDEE process-model (see details on ORCHIDEE in Section 2.2.2), to compare, for example, to -664 GgS.yr$^{-1}$ with the Simple Biosphere Model (SiB4 Kooijmans et al., 2021a). The oxic soils can both produce and consume COS but they are a net sink, estimated recently at -126 GgS.yr$^{-1}$ with the ORCHIDEE process-model (Abadie et al., 2022b). The net soil sink (taking into account the source due to anoxic soils and the net sink due to oxic soils) of COS is therefore estimated at -30 GgS.yr$^{-1}$ by Abadie et al. (2022b), to compare to, e.g., -89 GgS.yr$^{-1}$ according to SiB4 Kooijmans et al. (2021a).

– the atmospheric sink of COS is due to its oxidation by OH radicals and its photolysis in the stratosphere (estimated respectively at -130 to -80 GgS.yr$^{-1}$ and -50$\pm$15 GgS.yr$^{-1}$ by Whelan et al., 2018).

Reducing the uncertainties on the estimates of all these sources and sinks at the global scale at as fine temporal and spatial resolutions as would be required to bring information on $CO_2$ fluxes due to photosynthesis does not seem easy to accomplish in the next few years. The first challenge is the lack of knowledge still remaining on many processes which lead to COS or $CS_2$ emissions in the atmosphere. For example, Remaud et al. (2023a), Ma et al. (2023) and Ma et al. (2021) conclude that a source may be missing in the Tropics (probably from the ocean) and that the uptake of COS by vegetation at high northern latitudes is too small. Some natural processes emitting COS in the atmosphere are still only suspected, for example in plants used in agriculture (Belviso et al., 2022a; Maseyk et al., 2014; Bloem et al., 2012). The second difficulty is the lack of data on COS atmospheric mixing ratios (Montzka et al., 2007), which could be used in CTMs to evaluate the available estimations of sources and sinks at the regional scale, although the National Oceanic and Atmospheric Administration (NOAA) provides atmospheric mixing ratios of COS at several stations, which are useful at the global scale. Still, NOAA data are based on monthly or at best biweekly flask samples, limiting our ability to use them to constrain fluxes at scales smaller than the global to continental

scales. Observations with higher frequency are very sparse globally (Belviso et al., 2020; Kamezaki et al., 2023; Zanchetta et al., 2023; Kooijmans et al., 2016; Commane et al., 2015) and were not yet used for long-term systematic assessment of regional COS fluxes.

To begin to tackle the issue, Belviso et al. (2020) have made use of one of the few continuous time series of COS atmospheric mixing ratios, available in the Paris area, to assess the budget of COS in this area at the seasonal scale and during pollution peaks. Even though the area is almost always a COS sink, local biogenic emission episodes appear in summer and anthropogenic emissions from the Benelux, Eastern France and Germany are occasionally transported so as to influence the area in winter. Belviso et al. (2022b) then tried to use the information brought by the same COS time series of mixing ratios to

learn more about the anthropogenic emissions in the footprint of the station, which covers part of Western Europe (including Benelux, Eastern France and Germany) and part of the Atlantic. Anthropogenic emissions in the footprint of GIF were proven to be overestimated when analyzing specific events selected in the relatively long continuous time series available. Nevertheless, further characterization was not possible at the time of the study. Following these results, Belviso et al. (2023) used a semi-quantitative approach to assess the gridded inventory of direct and indirect anthropogenic emissions of COS by Zumkehr

et al. (2018) in France. The main conclusion of this work is that COS emissions in France are overestimated in this inventory by one order of magnitude and another way of mapping these emissions is required.

Therefore, this study aims at quantitatively assessing the anthropogenic and biogenic COS fluxes at the regional scale. It demonstrates that a set-up based on one measurement site in Western Europe, which provides data for over half a decade, makes it possible to:

– quantitatively assess the discrepancies in Zumkehr's anthropogenic emission inventory in the footprint of the measurement site

– evaluate a new inventory based on industrial emission declaration in the European Union

– study the seasonal and diurnal cycles of biogenic fluxes, based on the ORCHIDEE and SiB4 processed-based land surface models, and point to strengths and weaknesses in these two models.

For this, we use the continuous time series of COS mixing ratios measured in the Paris area from summer 2014 to the end of year 2019, as described in Section 2.1. We compare them to the concentrations simulated from marine, biogenic and anthropogenic fluxes in the area of interest (detailed in Section 2.2) combined to the contribution due to the rest of the world (Section 2.1) by the modeling tool described in Section 2.3. After an assessment of the general performances of the model (Section 3.1), we are able to quantitatively evaluate the anthropogenic sources from Western Europe as estimated by Zumkehr et al. (2018)

(hereafter referred to as "Zumkehr's inventory") and by our more targeted inventory (Section 2.2.3), confirming discrepancies from Belviso et al. (2023) in Zumkehr's inventory in France in particular, but also in Western Europe in general. Contrary to Belviso et al. (2023), the present study goes one step further by quantitatively assessing discrepancies in Zumkehr's inventory and by proposing a new inventory based on industrial emission declaration in the European Union. Having more reliable anthropogenic emissions, we can inquire into biogenic emissions, which is one of the main original purposes of studying COS.

We study the seasonal and diurnal cycles of biogenic fluxes, based on the ORCHIDEE and SiB4 processed-based land surface models (Section 3.3); this allows us to point to strengths and weaknesses in the two models.

## 2   Data and methods

### 2.1   Measurements and set-up of background mixing ratios

The measurements used in this study constitute a quasi-continuous time series of COS atmospheric mixing ratios obtained at
the monitoring site of Gif-sur-Yvette (GIF) located in the Paris area at 48.7109°N and 2.1476°E at 163 m asl with an inlet height 7 m agl. A commercial gas chromatograph (Varian 3800) coupled with a cryogenic preconcentrator (ENTECH P7100) for sample preparation, and a mass spectrometer detector (Varian Saturn 2200) for COS detection, were used to analyse this gas, as described by Belviso et al. (2013), Belviso et al. (2016) and Belviso et al. (2020). Calibration and drift correction is done every three weeks using a calibration gas containing 1.013 ppm of COS in helium, with occasional calibration using a standard
of compressed air with 573 ppt of COS, and another standard traceable to NOAA ESRL standard of 448.6 ppt. Calibrations led to a repeatability of 1 % and a precision of 0.2 %.

The GIF time series of hourly data spanning from August 2014 to December 2021 is available in Belviso et al. (2022b). In this study, we make use of the time series from August 2014 to December 2019 only (Figure 1, and Fig. A1 to A6), which covers the period of availability of inputs required for simulations, in particular global concentration fields used to compute the
background signal (see Section 2.1 and Section 2.3).

The time series of flask measurements sampled by the National Oceanic and Atmospheric Administration (NOAA) network at Mace Head (MHD) in Ireland at 53.3°N and 9.9°W at 42 m asl (described in Montzka et al., 2007, with 1 to 5 pairs of flask per month, collected mostly between 8:00 and 17:00 UTC) are also shown in Figure 1 to illustrate the so-called "background", i.e., the overall contribution of all the sources and sinks which are not in our area of interest. In our study, which focuses on
Western Europe and more particularly on the footprint of GIF, the mixing ratios at MHD are representative of the background when the air masses are advected from the West, which is a frequent meteorological configuration (Belviso et al., 2020).

The contribution of the background to the simulated mixing ratios is computed using three-dimensional fields at the global scale (see details in Section 2.3). The COS mixing ratio fields used here are obtained from Remaud et al. (2022), at a horizontal resolution of 3.75° in longitude and 1.875° in latitude for 39 pressure levels at a three-hourly time resolution (illustration in
Figure 2). These global atmospheric inversions were designed to assimilate data from the background NOAA observation sites, such as MHD.

### 2.2   Fluxes

In the following, "emission" denotes fluxes which are sources as seen from the atmosphere and "uptake" denotes fluxes which are sinks as seen from the atmosphere; "flux" is used for (ensembles of) processes which can be either sources or sinks.

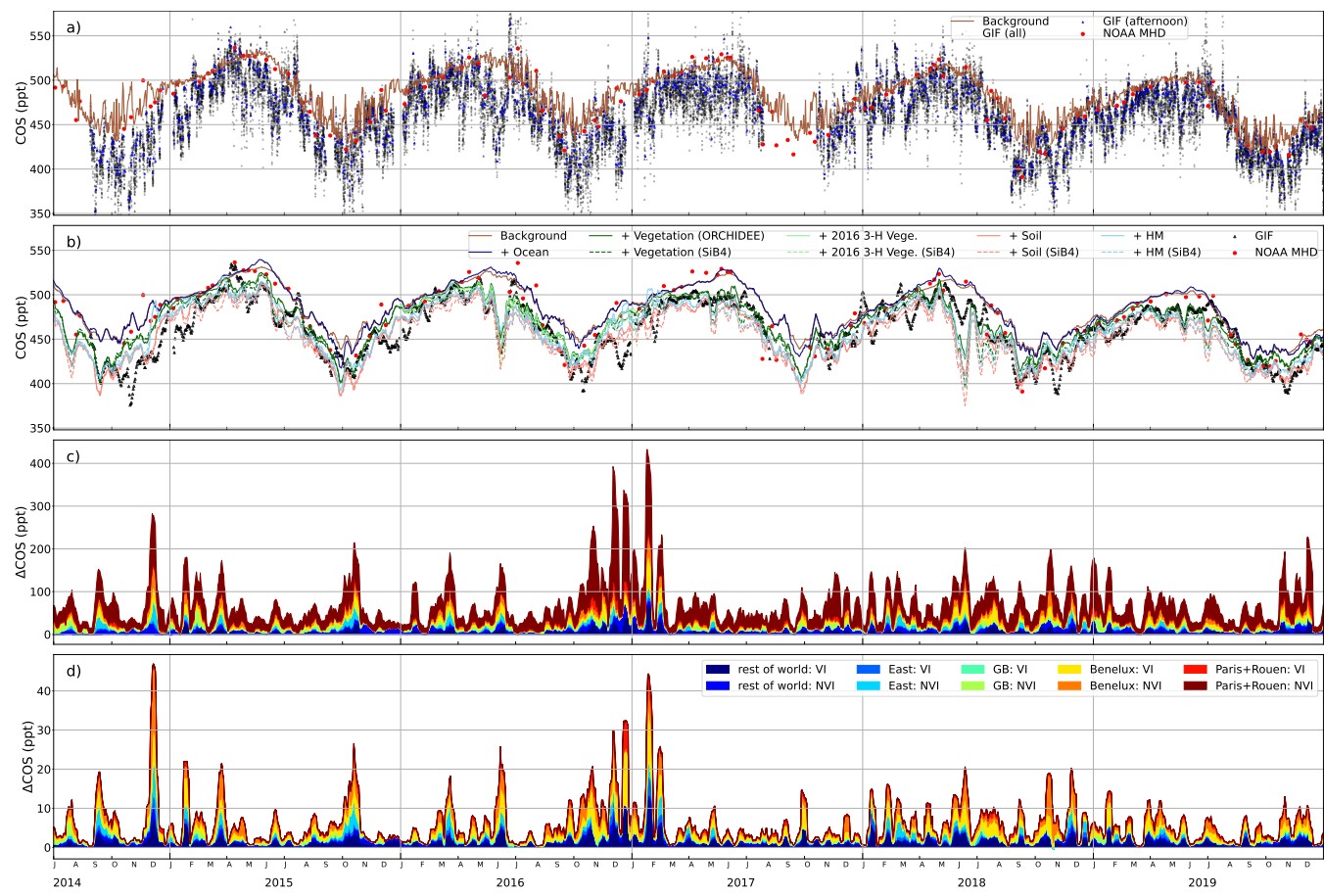

**Figure 1.** *Time series of observed and simulated atmospheric mixing ratios of COS. a) Hourly COS observations at GIF, with afternoon (12:00-18:00 UTC) averages super-imposed, and flask measurements at Mace Head (MHD) and simulated background described in Section 2.3. Remark: the limits set on the y-axis do not show ≈ 50 extremely low values and ≈ 190 extremely high values of hourly COS observations. b) 10-day rolling mean of GIF afternoon observations, with cumulative simulated contributions from background, ocean sources and biogenic land fluxes (see Section 2.3 for details, "2016 3-H Vege." = 3-hourly vegetation uptake as available for 2016), c) 10-day rolling mean of simulated anthropogenic contributions to COS mixing ratios by region (shown in Figure 2) and sector according to Zumkehr's inventory (see Section 2.2.3, VI = viscose industry, NVI = non-viscose-related emissions), d) 10-day rolling mean of simulated contributions by region (shown in Figure 2) and sector according to our home-made inventory ("HM", see Section 2.2.3, Figure 2). Remark: afternoon data are shown here because the model is assumed to better represent the vertical mixing in the afternoon so that the comparison to measurements highlights the discrepancies in fluxes compared to the model's errors.*

| Category | flux component | magnitude (GgS.y$^{-1}$) | references |
|---|---|---|---|
| natural oceanic | source via DMS and CS$_2$ | 265±210 | Lennartz et al. (2017, 2021) |
| | + emissions | 507 | Remaud et al. (2022) (see Sect. 2.2.1) |
| anthropogenic viscose industries | source via CS$_2$ | | |
| anthropogenic others | source via CS$_2$ | 400±180 | Zumkehr et al. (2018) |
| | + emissions | | |
| biomass burning | emissions | 60±37 | Stinecipher et al. (2019) |
| anoxic soils | emissions | 96 | Abadie et al. (2022a) |
| oxic soils | net sink | -126 | Abadie et al. (2022a) |
| net soils | net sink | -30 | Abadie et al. (2022a) |
| | | -89 | Kooijmans et al. (2021a) |
| vegetation | uptake | -530 | see Sect. 2.2.2 |
| | | -664 | Kooijmans et al. (2021a) |
| atmospheric oxidation by OH | sink | [-130 , -80] | Whelan et al. (2018) |
| atmospheric photolysis | sink | -50±15 | Whelan et al. (2018) |

**Table 1.** *COS sources and sinks: global estimates available at the time of this study, from Whelan et al. (2018); Remaud et al. (2023b).*

Our simulations, detailed in Section 2.3, take into account COS fluxes in the area of interest as provided by the data sets described here and listed in Table 1. The contributions from biomass burning emissions and the atmospheric sink are neglected in this study, as well as the emissions from anoxic soils. They play a significant role at the global scale for long-term studies (e.g., Ma et al., 2023), but are negligible compared to ocean emissions (see Sect. 2.2.1), biogenic fluxes (see Sect. 2.2.2) and anthropogenic emissions (see Sect. 2.2.3).

**2.2.1    Ocean emissions**

The ocean data set designed by Lennartz et al. (2021) and Lennartz et al. (2017) includes direct emissions of COS and indirect emissions of COS via CS$_2$ and dimethylsulfide (DMS) emissions at a monthly resolution and at a $1° \times 1°$ horizontal resolution, respectively. Anthropogenic DMS emissions also exist. However, Sarwar et al. (2023) and von Hobe et al. (2023) have shown that their impact (through oxidation) on simulated COS concentrations is negligible.

The emissions of the three species have been computed using coarse-resolution box models calibrated with ship-borne measurements made in different parts of the globe and result in a data-driven data set for COS oceanic emissions. This ocean data set has been optimized by Remaud et al. (2022) by assimilating NOAA flask data into the model LMDZ. This optimized data-set has been compared to others by Ma et al. (2023) (therein called "OPT-LMDZ") and gives very similar results to other datasets when evaluated with independent atmospheric data at the global scale. Our ocean fluxes give total (direct+indirect)
emissions of COS of 507 GgS.yr$^{-1}$ on average over 2014-2019 and 4 GgS.yr$^{-1}$ in our domain of interest.

### 2.2.2 Biogenic land fluxes

In this study, biogenic land fluxes refer to vegetation uptake and soil exchanges. We compare simulations based on the OR-CHIDEE (Krinner et al., 2005) and SiB4 (Haynes et al., 2019b, a) land surface models.

In ORCHIDEE and SiB4, a mechanistic representation of vegetation COS uptake has been implemented following Berry et al. (2013), and soil COS exchanges are computed based on the model from Ogée et al. (2016), representing both COS uptake and emission by soils. The representation of biogenic COS fluxes in SiB4 is described in details in Kooijmans et al. (2021b), while the vegetation and soil COS models in ORCHIDEE are presented in Maignan et al. (2021) and Abadie et al. (2022b), respectively.

Biogenic land fluxes are computed with a horizontal resolution of $1° \times 1°$ at a monthly time-step (illustrated in Figure 2), thus losing any temporal variability at the synoptic scale. The data set used here estimates the average uptakes of COS over 2014-2019 at $22.6 \, \mathrm{GgS.yr^{-1}}$ (resp. $29.8 \, \mathrm{GgS.yr^{-1}}$) by the vegetation according to ORCHIDEE (resp. SiB4) and $12.9 \, \mathrm{GgS.yr^{-1}}$ by the soil in our domain of interest. In particular, real-life biogenic fluxes exhibit a significant diurnal cycle (whereas modelled fluxes are constant) Indeed, vegetation COS uptake is regulated by stomatal conductance. There is a residual uptake during nighttime due to incomplete stomatal closure, and a stronger uptake during daytime when stomatal conductance increases. We assess the sensitivity of our simulations to daily varying biogenic fluxes by using 3-hourly vegetation uptake as simulated by ORCHIDEE and SiB4 for the year 2016. The different performances of the models with monthly and 3-hourly fluxes are evaluated in Sect. 3.3.

### 2.2.3 Anthropogenic emissions

Two different COS anthropogenic inventories are used in this study. The inventory by Zumkehr et al. (2018) accounts a total of $62.1 \, \mathrm{GgS.yr^{-1}}$ in the domain of interest, compared to our more realistic inventory, with a total of $11.2 \, \mathrm{GgS.yr^{-1}}$.

**Main characteristics of Zumkehr's inventory**

The inventory by Zumkehr et al. (2018) accounts for the sectors emitting the most COS and $CS_2$ at the country level and provides yearly emissions from 1980 to 2012. Here, the values for the year 2012, as the most recent available, have been used. In our domain of interest, they amount to a total of $62.1 \, \mathrm{GgS.yr^{-1}}$ among which $20.5 \, \mathrm{GgS.yr^{-1}}$ for the viscose industry, $14.5 \, \mathrm{GgS.yr^{-1}}$ for coal use, $23.6 \, \mathrm{GgS.yr^{-1}}$ for the pigments and $2.8 \, \mathrm{GgS.yr^{-1}}$ for aluminum production, with minor contributions from industrial solvents and the paper industry. The sub-country distribution is done according to secondary proxies, such as energy industry activity or industrial $CO_2$ emissions. This proxy-based distribution proved effective in the U.S., but can be misleading in some European countries. For example, in France, only one power plant is fueled by coal, and only a very few viscose sites are still active, not necessarily near the biggest cities in the country. On the opposite, Zumkehr's methodology leads to a distribution of national emissions around the main urban areas, decorrelated to real emissions. As shown in Figure 2 and Fig. 1c of Belviso et al. (2023), such a hot spot appears in the Paris area, with its center located close (about 20 km) to the North-East of GIF.

In the following, the sectors provided in Zumkehr's inventory (Zumkehr et al., 2018, therein Tab 1) are grouped as viscose industry emissions (abbreviated as "VI"), which include the sectors named "Pulp & Paper", "Rayon Staple", "Rayon Yarn", and non-viscose-related emissions (abbreviated as "NVI"), which include the sectors named "Agricultural Chemicals", "Aluminum Smelting", "Carbon Black", "Industrial Coal", "Residential Coal", "Industrial Solvents", "Titanium Dioxide (TiO2)", and "Tires". Maps and bar plots for selected European countries are available in Belviso et al. (2023) main text and supplementary material.

**Home-made inventory**

To overcome the caveats of Zumkehr's inventory in Western Europe, we designed our own inventory, following the conclusions of Belviso et al. (2023), and based on:

– explicit declaration of $CS_2$ emissions from the viscose industry (including rayon yarn and staple and other products such as cellulosic casings) with a comprehensive list of plants in Europe (abbreviated as "VI" and comparable to the VI emissions in Zumkehr's inventory) for the year 2018 (available at the time of the study); see details in Tables 2-3 of Belviso et al. (2023). The total emissions in our domain of interest are $\approx 4.7\,GgS.yr^{-1}$ in 2018. How our simulations take into account the oxidation of $CS_2$ into COS is described in Section 2.3.

– for coal, $CO_2$ emissions for all coal power plants in Europe (from the Global Energy Monitor's Global Coal Plant Tracker) were used to estimate COS emissions for each year based on a unique emission factor (abbreviated as "NVI" and comparable to the NVI coal-related emissions in Zumkehr's inventory). We used a value of $5.7 \times 10^{-6}$ molecules of COS emitted for each molecule of $CO_2$ during coal combustion. The average emissions over 2014-2019 are $\approx 6.5\,GgS.yr^{-1}$ in our domain of interest.

Through lack of data, we did not apply any temporal profiles to viscose and coal emissions, even though they are expected to strongly vary over the year, even from one week to the next one. For viscose, we applied no inter-annual variability and keep emissions stable based on the year 2018. For coal emissions, inter-annual variability is based on yearly $CO_2$ emission values. Compared to Zumkehr's inventory, ours (Figure 2) displays only a few hot spots of emissions in France: 4 viscose industry sites with big factories and 4 coal-burning power plants. The closer to GIF (large black triangle in Northern France in Figure 2c) is in the city of Beauvais, which lies about 85 km to the North of GIF, i.e., further than the Paris area.

## 2.3 Simulations of concentrations

The modelled COS mixing ratios were obtained by using the Lagrangian atmospheric transport model FLEXPART (FLEXible PARTicle) version 10.3 (Stohl et al., 2005; Pisso et al., 2019). The model FLEXPART is driven by meteorological data from the European Centre for Medium-range Weather Forecast (ECMWF) ERA5 (Hersbach et al., 2020) with 3-hourly intervals and 60 vertical layers, retrieved using the FLEX-extract toolbox (Tipka et al., 2020). The performances of ERA5-provided boundary layer height and wind direction are very good (e.g. Molina et al., 2021). In our case, meteorological data is provided to FLEXPART at a $1°$ horizontal resolution. For the whole duration of the observation period, 2000 virtual particles are

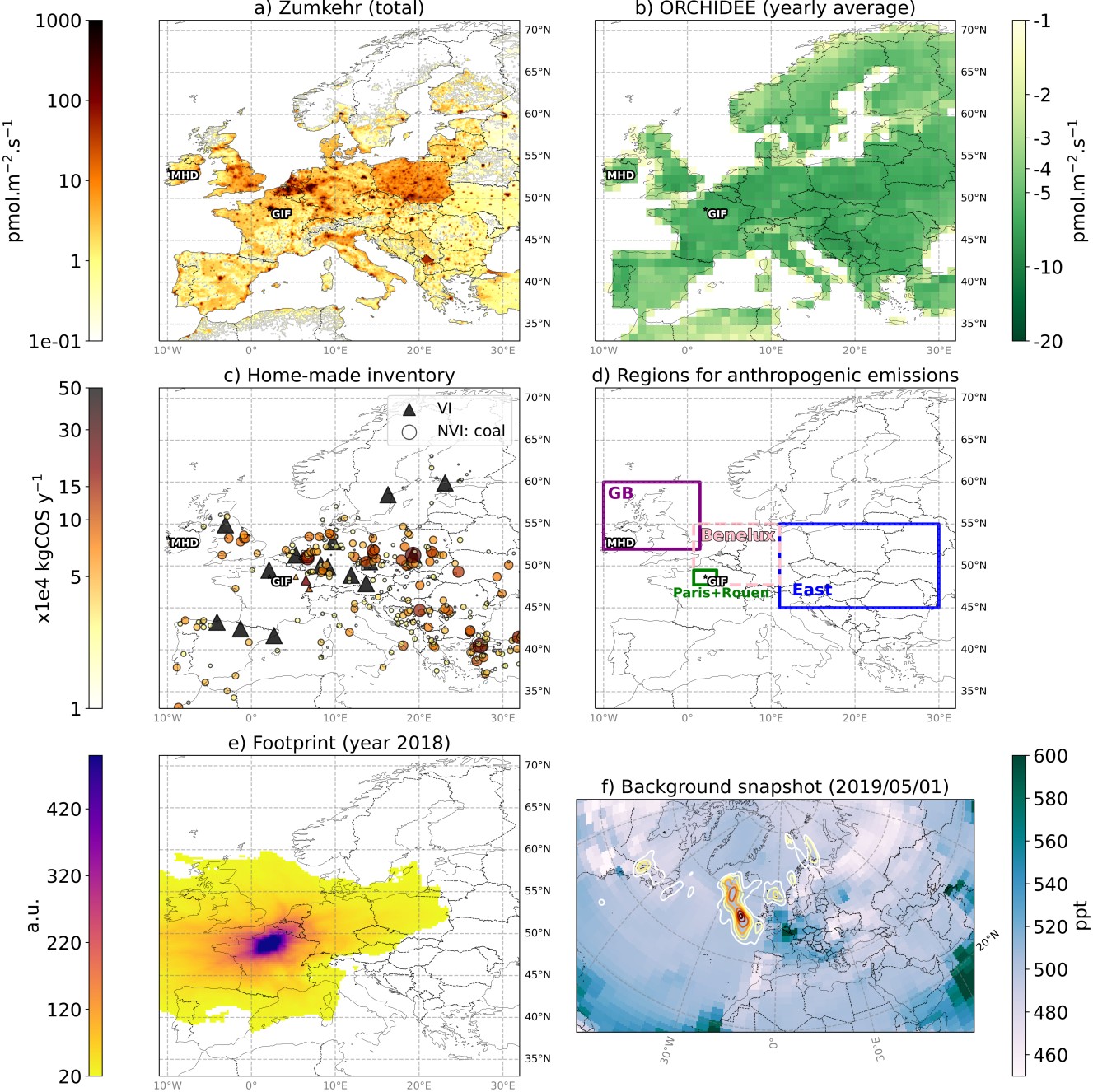

**Figure 2.** *Maps of input data sets used in our study. a) Zumkehr's yearly, i.e., 2012 total emissions, b) 2019 yearly average biogenic land fluxes from the ORCHIDEE model, c) emission points from viscose and coal industries as defined using our methodology (see Section 2.2.3 for details, here year 2019 for illustration), d) regions of interest for anthropogenic emissions, e) illustration of footprint summed-up over the year 2018, f) snapshot of background concentration fields; contours highlight the computed sensitivity to the background 10-days before observation at GIF (see Sect. 2.3 for details). Remark: units for area (gridded) emissions are $pmol.m^{-2}$ whereas emissions by point sources are in mass units; footprints are shown in arbitrary units (a.u.).*

| Fluxes from / Region | ocean | biogenic land | Zumkehr's VI | Zumkehr's NVI | home-made VI | home-made NVI |
|---|---|---|---|---|---|---|
| whole area of interest | o | bl | | | | |
| Benelux ("B") | | | ZviB | ZnviB | HMviB | HMnviB |
| Paris+Rouen area ("U") | | | ZviU | ZnviU | HMviU | HMnviU |
| area of interest excluding Benelux and the Paris+Rouen area | | | ZviW | ZnviW | HMviW | HMnviW |

**Table 2.** *FLEXPART simulations per region (shown in Figure 2) and sector. Each cell: a group of letters indicate that the simulation is run; the same IDs are used for each violin plot in Figure 3. "Z" = Zumkehr's inventory, "HM" = home-made inventory (this study), "VI" or "vi" = viscose industry, "NVI" or "nvi" = non-viscose-related (actually coal-related only in HM, see Section 2.2.3), "U" = Paris+Rouen area = an urban and industrial area, "B" = Benelux, "W" = whole area excluding "B" and "U".*

released every 6 h and followed backward in time for 10 days. The multiple FLEXPART simulations are driven by the GUI-based toolbox designed by Berchet et al. (2023). FLEXPART and ERA5 meteorological data are well recognized tools in the atmospheric community, the combination of the two being widely used in the atmospheric inversion community (Pisso et al., 2019; Bakels et al., 2024). The choices made for FLEXPART's configuration (number of particles released, frequency of releases, resolution) are consistent with the set ups used for the atmospheric inversions fluxes (e.g. Bergamaschi et al., 2022; Thompson et al., 2021).

We assume a conversion of $CS_2$ to COS with a maximum of 87% molar conversion rate, as used by Zumkehr et al. (2018). The kinetics of the conversion is approximated by prescribing a half-life of 3 days to $CS_2$ in the atmosphere. The amount of COS created from a $CS_2$ source along the transport path of the air mass is then given as $0.87 \times \left(1 - e^{-\frac{t}{\frac{\tau}{\ln 2}}}\right)$ with $\tau = 3$ days. Note that a life time of 1.5 day has also been tested (not shown), with almost no changes in the conclusions of this study.

So-called source-sensitivity (or footprint) maps (Seibert and Frank, 2004) are computed for every release date by counting the number of particles transported above a certain point below a threshold of 500 m above ground level. Source contribution by process-type are inferred by convolving source-sensitivity maps with flux maps from the ocean emissions, the biogenic fluxes and one of the two anthropogenic emission inventories (see Section 2.2.1, Section 2.2.2, Section 2.2.3). One FLEXPART simulation is run for each process or sector and region of interest (see Figure 2) so that the simulations (Table 2) can be combined to include or exclude particular sectors and/or particular regions when building the footprint maps.

The background mixing ratios (Fig. 2f) are calculated by combining the 3D source-sensitivity fields (e.g., Thompson and Stohl, 2014; Pisso et al., 2019) at the end of the backward trajectories with the available COS concentration field (see Section 2.1). The background thus obtained represents the average of the mixing ratios in the grid cells where each particle trajectory terminated 10 days before the observation. For instance, as illustrated in Fig. 2f, for the given date, GIF observations are sensitive to background mostly in North-Western Atlantic (higher sensitivity for deep red contours). The computation of

COS contributions from surface fluxes and the background was carried out using the Community Inversion Framework (CIF; Berchet et al., 2021).

## 3 Results and discussion

### 3.1 General patterns of simulated and measured COS concentrations at GIF

At GIF, the seasonal variations of COS mixing ratios are dominated by the contributions of the background and ocean i.e. by large scale fluxes; the variations at shorter time scales (week or day) are driven by the biogenic land contribution (Belviso et al., 2023). Finally, depending on the wind speed and direction, anthropogenic emissions may dominate for short episodes of high concentrations (see section 3.3.2 Selected winter episodes in Belviso et al., 2023). The contributions to COS mixing ratios at GIF due to the ocean, the biogenic land fluxes and the anthropogenic emissions are shown in Figure B1.

In the following, we use the afternoon (12:00-18:00 UTC) daily averages of measured and simulated mixing ratios because the model is assumed to better represent the vertical mixing in the afternoon so that the comparison to measurements highlights the discrepancies in fluxes compared to the model's errors. More detailed results for the whole day or nighttime and per season are shown in Table C1.

By design, the background contribution (Section 2.1) fits the main monthly variations as measured at a background station 265 such as MHD (Figure 1 a) and b)). Its contribution at GIF ensures a good simulation of the variability (Table 3: Pearson's correlation $\geq$ 0.75) and a base-line mean error $\leq$ 35 ppt over the whole period of interest. As expected, adding the natural emissions from the ocean (Section 2.2.1) and the biogenic land fluxes (Section 2.2.2) reduces the bias (by almost 15 ppt in absolute value) and the mean error is decreased by more than 20%.

As mentioned in Sect. 2.2.2, we assess the impact of using temporally resolved biogenic fluxes on our simulations. Making 270 use of a vegetation uptake with a 3-hourly time resolution (during only the year 2016, see Section 2.2.2) does not improve the statistical fit of the model to the observations: the three statistical indicators are almost unchanged (Table 3: "period" = 2016). The variability is not better reproduced when adding the natural emissions from the ocean and the biogenic land emissions from the soils and the vegetation to the background (correlations $\leq$0.75 in all cases). This may be due to the monthly resolution of the ocean emissions (Section 2.2.1) being too coarse compared to the variability of the transport and to an issue with the 275 variability of the vegetation uptake or the soil exchanges, probably at the seasonal scale, which is assessed in Section 3.3.

The limited impact, and even degradation, of performances when using the diurnal cycle of biogenic flux suggests an issue in their representation in ORCHIDEE, maybe in the residual ecosystem (vegetation and soil) COS flux simulated at night.

The contributions of the anthropogenic emissions by Zumkehr degrades all three indicators of the fit to the measured concentrations at GIF: the bias and mean error are very high (both $\geq$100 ppt), the variability is not reproduced anymore. The regional 280 anthropogenic emissions located in the Paris+Rouen area explain the major part of the discrepancies between the simulation and the measurements (Table 3). According to Zumkehr's inventory, COS emissions in Île-de-France, i.e., the Paris area itself, are mostly ($\approx$62%) due to viscose industry, which is not consistent with the lack of any such factory declaring emissions in this region. The small (<1%) contribution by coal-using industry is not consistent with the last coal power plant in the area being

closed in 2012. Maybe due to the same type of incomplete information or lack of relevant proxy, the overall European COS
emissions in Zumkehr's inventory may be overestimated, as suggested by the very high contributions (more than 100 ppt) due
to anthropogenic emissions alone simulated at GIF (Figure 1c).

The contribution of the anthropogenic emissions from our home-made inventory increases the bias and mean error compared
to the contributions of the background and natural fluxes only (Table 3). The maximum simulated contributions are between 40
and 50 ppt for some months (Figure 1 c and d), i.e., 10 times smaller than with Zumkehr's inventory. Therefore, the bias and
mean error computed over the whole period using our inventory remain more than twice smaller than with Zumkehr without the
emissions of the Paris+Rouen area. This order of magnitude is more consistent with the expected anthropogenic contribution
required to match the measured concentrations above the background at GIF. Overall statistics are very similar when using
only viscose or coal, or both. Further investigation is needed to narrow the range of anthropogenic fluxes in Europe. However,
as anthropogenic contributions are observed as peaks, the magnitude of which is difficult to reproduce in transport models, a
combination of several observation sites would be needed.

The information which can be retrieved from the time series of concentrations in one measurement site on the relevancy of
our inventory compared to Zumkehr's is discussed in the following in terms of activity sectors and source regions.

### 3.2    Anthropogenic emissions

The general performances of the model at GIF (Section 3.1) show that the total emissions of Zumkehr's inventory lead to a
large overestimation of COS concentrations at GIF (Table 3), almost half of which is due to the emissions located in the areas
of Paris and Rouen. We represent in Figure 3 the distributions of observations and simulations from different sectors as violin
plots. Above the overestimation due to the "natural" contributions, Zumkehr's inventory leads to a median overestimation of
$\approx$52 ppt (Figure 3, "Z total" vs "natural"). The non-viscose-related emissions in the Paris and Rouen areas explain almost half
of this discrepancy (24 ppt, "Z total" vs "ZviW + ZnviW + ZviB + ZnviB + ZviU"). The non-viscose-related emissions in "the
rest of the world", i.e., the whole world excluding the Paris and Rouen areas and the Benelux explain a bit less than a quarter
of the overestimation (12 ppt, "ZviW + ZnviW" vs "ZviW" vs "natural"). Of the remaining 12 ppt, 8 ppt are due to viscose
industry emissions in the rest of the world and 1.5 ppt to viscose industry emissions in the Benelux alone. The unrealistic
COS emissions in the Paris and Rouen areas not only lead to too high median contributions but also to a large number of high
concentration peaks, as shown by the upper elongation of the violin "Z total" (maximum actually $> 1995$ ppt). As expected, our
home-made inventory is more compatible with the observations in GIF (Figure 3). In this case, the background and the oceanic
and biogenic land fluxes contribute for $\approx$40% to the median overestimation ($\approx$4 ppt, Figure 3: "GIF obs" vs "natural") of the
observations, compared to 60% for the home-made anthropogenic emissions ($\approx$7 ppt, Figure 3: "HM total" vs "natural"). The
distribution of extreme events, i.e., with very high or very low COS concentrations obtained with our home-made inventory is
closer to the observed one (Figure 3, upper and lower elongations of "GIF obs" vs "Z total" vs "HM total").

Additional continuous observation sites would be needed in different places of Europe to clarify the relevancy of our in-
ventory beyond the Paris area and vicinity. In addition, our inventory is based on coarse emission factors, both for the viscose

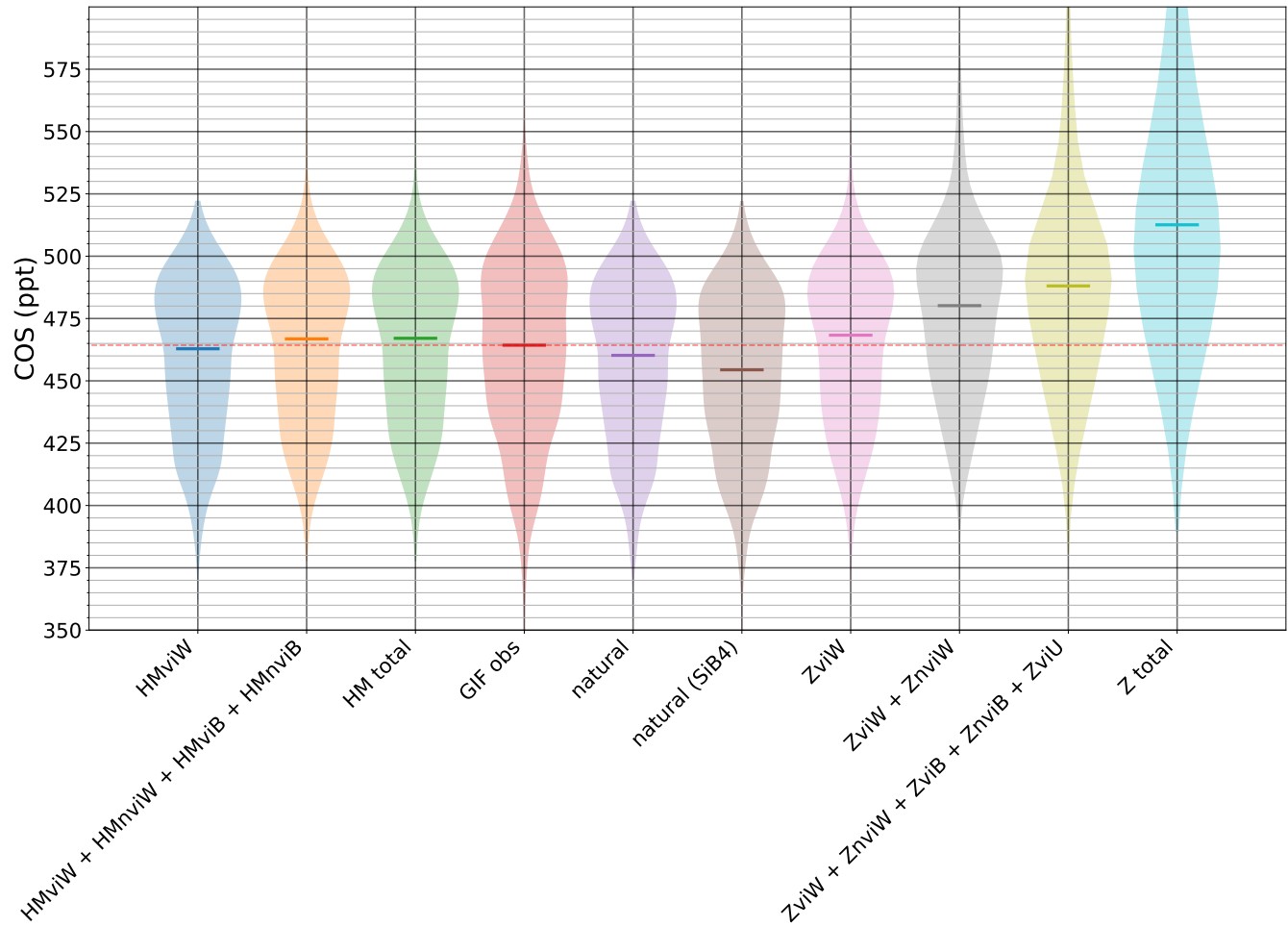

**Figure 3.** *Violin plots of COS mixing ratios at GIF (solid horizontal lines show the medians) for observations ("GIF obs") and various simulations taking into account different sets of contributions. See Table 2 for the available simulations per region and per sector; "natural" = background + o + bl; "Z total" = natural + ZviW + ZnviW + ZviB + ZnviB + ZviU + ZnviU; "HM total" = natural + HMviW + HMnviW + HMviB + HMnviB + HMviU + HMnviU.*

| Contribution | period | bias (ppt) | RMS (ppt) | corr |
|---|---|---|---|---|
| Background | whole | 23 | 33 | 0.76 |
| Background + ocean | whole | 23 | 33 | 0.75 |
| Background + biogenic land | whole | -5 | 25 | 0.73 |
| Background + biogenic land | 2016 | -4 | 25 | 0.72 |
| Background + biogenic land with 2016 3-hourly vegetation | 2016 | -5 | 26 | 0.71 |
| Background + ocean + biogenic land | whole | -5 | 25 | 0.74 |
| Background + ocean + biogenic land | 2016 | -3 | 25 | 0.73 |
| Background + ocean + biogenic land with 2016 3-hourly vegetation | 2016 | -4 | 26 | 0.72 |
| Background + anthro. Zumkehr | whole | 100 | 151 | 0.11 |
| Background + anthro. Zumkehr w/o Paris/Rouen | whole | 58 | 87 | 0.27 |
| Background + coal (HM) | whole | 26 | 35 | 0.73 |
| Background + viscose (HM) | whole | 25 | 34 | 0.74 |
| Background + coal (HM) + viscose (HM) | whole | 27 | 37 | 0.71 |
| Background + ocean + biogenic land + anthro. Zumkehr | whole | 72 | 129 | 0.14 |
| Background + ocean + biogenic land + anthro. Zumkehr w/o Paris/Rouen | whole | 29 | 66 | 0.33 |
| Background + ocean + biogenic land + coal HM + viscose HM | whole | -1 | 25 | 0.74 |

**Table 3.** *Statistical indicators of the performances of the model compared to the measurements at GIF for each contribution, based on the daily afternoon (12:00-18:00 UTC) means of simulated and measured mixing ratios; indicators are either computed over the whole period, i.e., Aug. 2014 - Dec. 2019 (see Section 2.1) or only over the year 2016 for comparison with the available contribution by the 3-hourly varying vegetation. Bias = mean difference model minus measurement; RMS = root mean square model minus measurements; corr = Pearson's correlation coefficient between model and measurement time series. HM indicates emissions from our home-made inventory (see Section 2.2.3).*

industry and for coal power plant. Facility-level campaign as carried out around a viscose factory near Rouen in Belviso et al. (2023) would significantly improve our understanding of COS anthropogenic fluxes, thus our inventory.

### 3.3 Biogenic land fluxes

Disentangling the main contribution of mismatches between observations and simulations is very challenging with one site only. Differences at the synoptic scale, as represented in Figure 1, can originate from erroneous background, transport discrepancies, or from incorrect fluxes. Moreover, with one site only, absolute values of the biogenic fluxes cannot be systematically estimated. We therefore focus on the variations through time, particularly between day and night and seasonally. As represented in the time series in Appendix A, fine scale temporal variability, corresponding to the diurnal cycle, is well reproduced during

some periods of the year, especially in spring. The diurnal cycle is dominated by local influences, with remote influence of the background and of distant fluxes transported to GIF filtered out. We therefore focus here on 12-hour day or 12-hour night differences in measured or simulated COS concentrations, defined as either "daytime difference", hereafter $\Delta_{day}^{COS}$, between day D

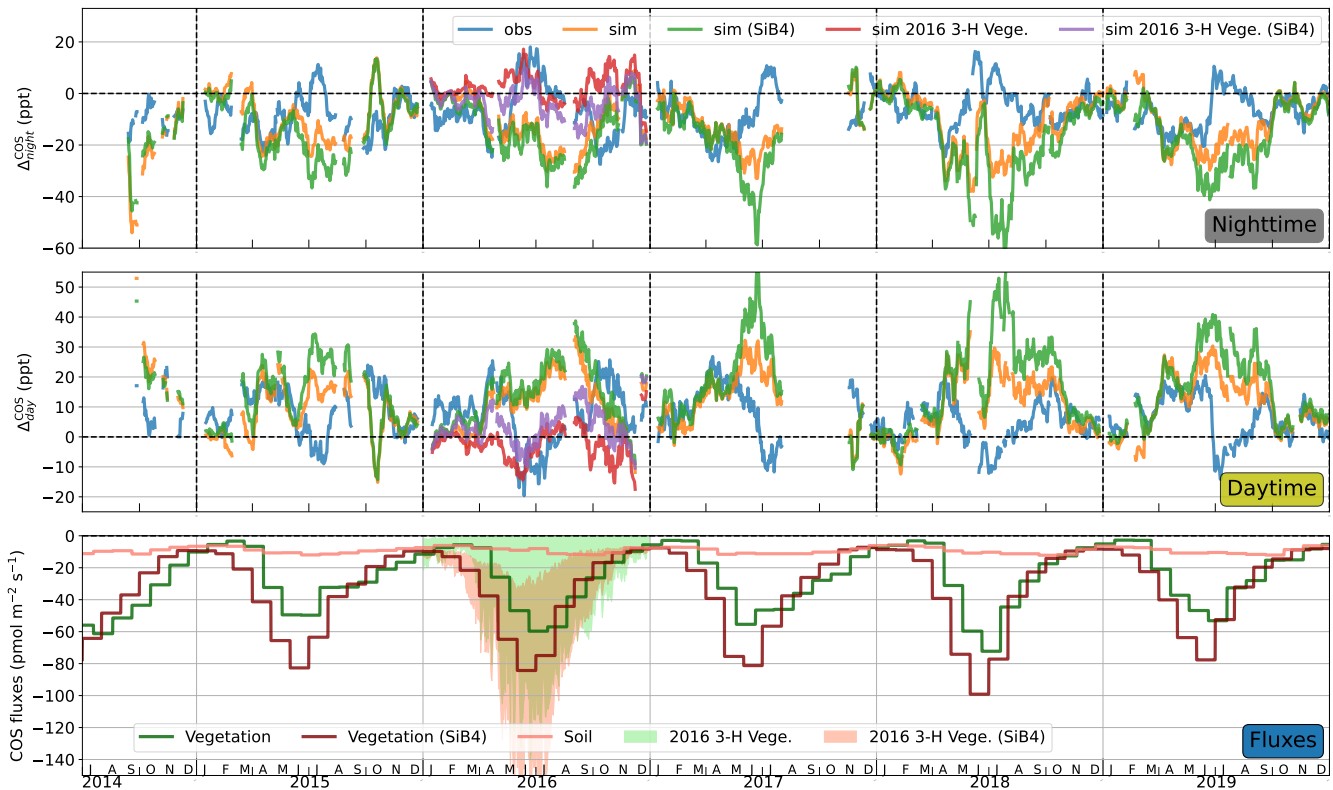

**Figure 4.** *15-day rolling mean of the nighttime ($\Delta_{night}^{COS}$, top) and daytime ($\Delta_{day}^{COS}$, middle) differences of the observed and simulated COS concentrations at GIF with matching COS total uptake by vegetation and soils in the domain of interest, using the models ORCHIDEE and SiB4 (see Section 2.2 for details). Nighttime difference = $\Delta_{night}^{COS}$ = $[COS]_{D@06:00} - [COS]_{D-1@018:00}$, daytime difference = $\Delta_{day}^{COS}$ = $[COS]_{D@18:00} - [COS]_{D@06:00}$ (see text for more details); contributions taken into account in the simulations are the background + ocean + our home-made anthropogenic inventory + either biogenic land with a monthly time-resolution ("sim") or biogenic land with 2016 3-hourly time-resolution for vegetation ("sim 2016 3-H Vege."); fluxes are uptakes by the soil, the vegetation at a monthly ("Vegetation") or 3-hourly ("2016 3-H Vege.") resolution (see Section 2.2 for details).*

at 6 p.m. and day D at 6 a.m. or "nighttime difference", hereafter $\Delta_{night}^{COS}$, between day D+1 at 6 a.m. and day D at 6 p.m.. Thus, these differences are mostly influenced by a combination of small scale meteorological conditions (mostly the diurnal cycle of

330 the planetary boundary layer) and of regional fluxes (within the transport footprint around the station). In the present analysis, we assume that all discrepancies are attributable to fluxes, even though the diurnal cycle of transport in FLEXPART is imperfect. Discrepancies in the diurnal cycle of transport patterns would only impact the magnitude of the day/night differences, but not the overall patterns discussed below, thus not impacting our qualitative conclusions.

The variations of the $\Delta_{day}^{COS}$ and $\Delta_{night}^{COS}$ over the available 5.5 years of observation are represented in Figure 4, alongside COS

fluxes from regional vegetation and soils, using both the ORCHIDEE and SiB4 models. The observed seasonal cycle of $\Delta_{night}^{COS}$ and $\Delta_{day}^{COS}$ is similar for all years. The observed $\Delta_{night}^{COS}$ are slightly negative in winter (JFM; $-7$ ppt) then strongly negative

in spring (AMJ; $-10$ ppt) with yearly peaks of $\approx -20$ ppt; then, summer positive peaks of 10–15 ppt in July are followed by another negative low in fall ($-8$ ppt). The positive $\Delta_{night}^{COS}$ in summer are most probably due to emissions by the crops, such as rapeseed (Belviso et al., 2022a). Observed $\Delta_{day}^{COS}$ are opposite with slightly different magnitudes.

In winter (JFM), the vegetation is mostly inactive, and soils are at their lowest sink activity with the net soil uptake becoming larger than the vegetation uptake. Thus the contribution of the uptake by the vegetation and soils in our simulations in both ORCHIDEE and SiB4 for both time-resolutions (monthly or 3-hourly) is very small. Both the simulated $\Delta_{night}^{COS}$ and $\Delta_{day}^{COS}$ are therefore close to zero (on average, $-2$ ppt for nighttime and $+1$ ppt for daytime), contrary to observed $\Delta_{night}^{COS}$ and $\Delta_{day}^{COS}$ ($-7$ ppt for nighttime and $+7$ ppt for daytime). This suggests under-estimated COS uptakes at night in winter in ORCHIDEE

and SiB4 vegetation and/or soil datasets.

     During spring (AMJ), vegetation COS uptake is activated in ORCHIDEE and SiB4. When using constant monthly fluxes with no diurnal cycle, their contribution to the simulated nighttime uptake leads to lower concentrations at night compared to daytime, consistent with observations. In contrast, daytime concentrations are the result of compensating vegetation uptake and mixing in the boundary layer. However, the use of 3-hourly vegetation uptake by ORCHIDEE daily cycle results in

simulated $\Delta_{night}^{COS}$ and $\Delta_{day}^{COS}$ close to zero, not consistent with the observations, most likely related to absent uptake at night in ORCHIDEE. In contrary, in SiB4, persistent uptake occurs at night, leading to more realistic simulation than with ORCHIDEE. This discrepancy suggests a persistent nighttime uptake not adequately represented by ORCHIDEE's diurnal cycle. The OR-CHIDEE minimal stomatal conductance at night (see Section 2.2.2) has to be revised. Refining the model ORCHIDEE to improve the representation of nighttime processes may enhance its ability to reproduce observed dynamics, compared to SiB4.

Summer (JAS) shows opposite behavior compared to spring, with positive nighttime enhancements. These enhancements are not properly reproduced by both ORCHIDEE and SiB4 when using constant monthly fluxes. With 3-hourly fluxes, both models perform well in summer. Using a realistic diurnal cycle is key to reproducing the positive $\Delta_{night}^{COS}$ and negative $\Delta_{day}^{COS}$. The magnitude of fluxes in ORCHIDEE seems to be in better agreement with observations than with SiB4.

     During fall (OND), the performance of our simulations exhibits variability across different years, regardless of the inclusion

of a vegetation diurnal cycle. Notably, the simulations for the years 2016 and 2017 show significant discrepancies, whereas the results for the remaining four years align well with observational data. This disparity underscores the challenges associated with accurately reproducing the timing of vegetation senescence, particularly given the influence of intricate synoptic meteorological conditions on both vegetation dynamics and the performance of the transport model. The complexity of these interactions poses a significant hurdle in achieving consistent simulation outcomes across all years.

**4   Conclusions**

The present study analyzes 5.5 years of quasi-continuous COS measurements from the site GIF in the Paris region. At GIF, the seasonal variations of COS mixing ratios are dominated by the contributions of the background and ocean; the weekly to daily variations are driven by the biogenic land contribution; anthropogenic emissions may dominate for short episodes of high concentrations (Belviso et al., 2023). Through systematic comparison of measurements with simulations using backward tra-

370 jectories computed with the Lagrangian model FLEXPART, we provide a quantitative assessment of natural and anthropogenic COS fluxes in Western Europe.

Regarding industrial-emitted COS, we highlight the unrealistic magnitude of anthropogenic fluxes as provided by Zumkehr's inventory (Zumkehr et al., 2018) in Europe. In particular, Zumkehr's inventory suggests very high emissions in the Paris area, inconsistent with the absence of any coal power plant and major viscose industry in the region. We propose another inventory

based on reported industrial emissions as well as coal power plants in Europe. The new inventory proves much more consistent with observations and includes only a limited number of emission hotspots in France. Still our inventory comes with several limitations. First, it takes into account the emissions as reported by the "industries"; this category includes only sources above a legally binding threshold. Therefore, we do not account for sources which are under the threshold and we assume them to be small compared to the reported ones; but their overall magnitude remains actually unknown. Second, another limitation

arises from the flat temporal variability applied to anthropogenic fluxes in our inventory (similarly to Zumkehr et al., 2018) due to the lack of information on industrial and power-generation activity in viscose factories and coal power plants. This is expected to be a critical limiting factors as coal power plant activity depends on energy demands, and viscose industry activity are often organized by batches with days of intense emissions followed by periods with limited emissions. Third, we apply a single emission factor between $CO_2$ and COS emissions from coal power plants, whereas this factor depends on the coal used

in the combustion, as well as the overall properties and technologies used in the power plant. Last, limitations are also due to the model-based approach of our study. The sources of COS or its precursors are stacks of factories or coal power plants, i.e., hot-spots: high sources localized in a very small area compared to the resolution of the chemistry-transport model used to simulate the concentrations. Therefore, the plumes emitted from these hot-spots are not always well represented by the model, depending on the meteorological situation. For hot-spots close to the measurement site (e.g., from the Beauvais region 50 km

North of GIF where viscose factories are active), the numerical diffusion of the plumes of COS represented in the model leads most of the time to underestimating simulated concentrations; in particular events, if the error on the wind direction is large enough, the plume may be unduly directed to the site in the model, which leads to overestimating the simulated concentrations. Overcoming the above-mentioned limitations in assessing anthropogenic emissions would mostly need additional observation sites in the region (e.g., Zanchetta et al., 2023, in the Netherlands) to cross-validate our results.

For anthropogenic and oceanic $CS_2$ emissions, the kinetics of the conversion of $CS_2$ into COS through its ts oxidation by OH is not well-known. Moreover, through OH availability and temperature, the lifetime of $CS_2$ depends on the season and has a diurnal cycle. The impacts of these variations on the chemical source of COS must be assessed, when more relevant information, e.g. varying diurnal emissions are available.

Regarding soil and vegetation fluxes, we are able to suggest that the net ecosystem COS uptake simulated by both OR-

400 CHIDEE and SiB4 is underestimated in winter at night, requiring improvements in the nighttime processes in land-process models. The uncertainties in the modeling of transport at night in winter prevent us from quantifying the under-estimation of the nighttime COS uptake. However, the discrepancies between simulations and observations shown by the nighttime and day differences suggest the existence of such an under-estimation. Even when using ORCHIDEE's vegetation and soil uptakes with a diurnal cycle at a 3-hourly resolution, the analysis of the simulated diurnal cycle of COS concentrations confirms that a

night-time uptake is not adequately represented by ORCHIDEE, maybe due to the parameterization of the nighttime uptake by plants for COS. On the opposite, SiB4 has a more realistic representation of spring diurnal cycles of fluxes, leading to better agreement between simulations and observations of concentrations. In summer, the magnitude of fluxes by ORCHIDEE is more consistent than SiB4 fluxes when comparing to observations. Further work is needed to properly compare ORCHIDEE and SiB4 using day/night differences as the model FLEXPART is imperfect in the representation of the diurnal cycle of transport and dispersion. An analysis discriminating between the impacts of i) the fluxes, ii) the chemical source, iii) the vertical transport and iv) the advection on COS mixing ratios could be performed from a set of simulations, with a careful design, computer resources and up-to-date hourly varying fluxes. Replicating our study with another transport model and other meteorological constraints would allow to assess the uncertainty on diurnal simulations, even these uncertainties would only impact the magnitudes of the simulations and not their patterns, hence leading to unchanged conclusions.

Although it is based on only one site, the present study provides sufficient elements to identify discrepancies in COS emission data bases and land-process models in Western Europe, due to incorrect parameterizations of fluxes, or missing processes. Still, our single-site study is not sufficient to overcome those discrepancies to reach the level of precision needed to use COS observations as a proxy of plant uptake. A full network of observation sites, analog to the Integrated Carbon Observatory System (ICOS; Ramonet et al., 2011), would be a requirement to better quantify anthropogenic fluxes and thus to fully use COS for assessing $CO_2$ uptake by photosynthesis.

*Code and data availability.* GIF data is available from https://doi.org/10.14768/6800b065-dcec-4006-ada5-b5f62a4bb832. The CMIP6 version of the ORCHIDEE model including the vegetation and soil COS submodels is available upon request to the authors. Optimized COS concentration fields at the global scale by Remaud et al. (2023a) are available at https://zenodo.org/records/7632737 (OPT-LSCE).

*Author contributions.* AB designed and ran the simulations, including formatting of the home-made emission inventory and of observations; IP and AB analyzed the results, based on preliminary work by CH; CN and CH computed the FLEXPART simulations on which the analysis is based; MR provided the global COS inversions used to determine the background; SB provided the observations at GIF and the home-made French inventory of point sources; CA and FM provided ORCHIDEE fluxes; all co-authors contributed to writing the text.

*Competing interests.* The authors declare no competing interests relative to the present study.

*Acknowledgements.* This research has been supported by the 4C project of the European Commission's Horizon 2020 framework programme (Grant 821003). This study was partially funded by the CO2 Human Emissions (CHE) project, which received funding from the European Union's Horizon 2020 research and innovation programme under Grant 776186. Calculations were performed using the resources of LSCE, maintained by J. Bruna and the LSCE IT team.

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

## Appendix A: Time series of observations and simulations

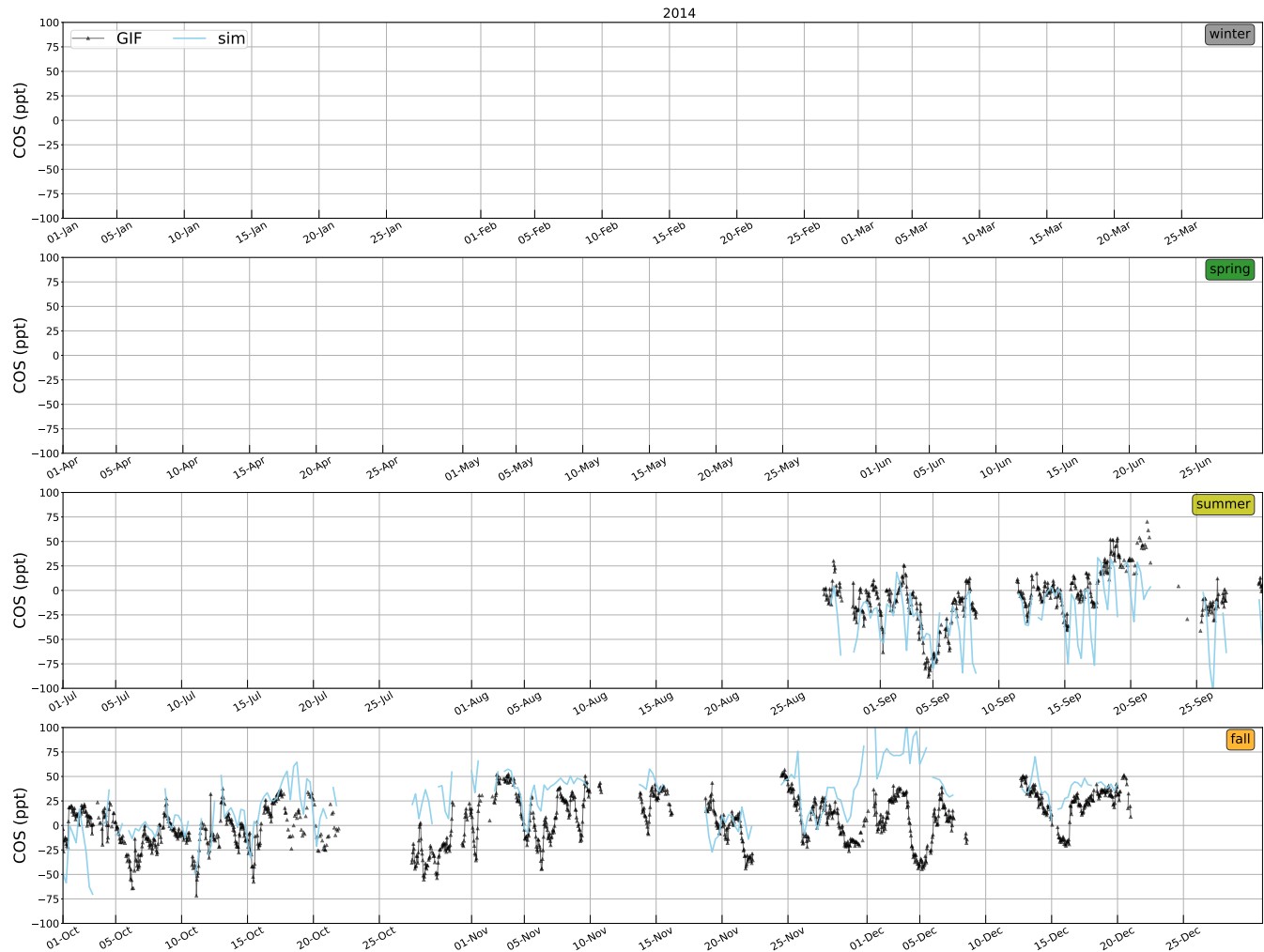

**Figure A1.** *Smoothed time series of observed and simulated atmospheric mixing ratios of COS at GIF in 2014. Observations are hourly averages; contributions taken into account in the simulations are the background + ocean + our home-made anthropogenic inventory + biogenic land with a monthly time-resolution for vegetation and soils (see Section 2.2 for details); smoothed by substracting the rolling 20-day mean of the observations to the observed and simulated time series. Note that the time series begins end of August 2014.*

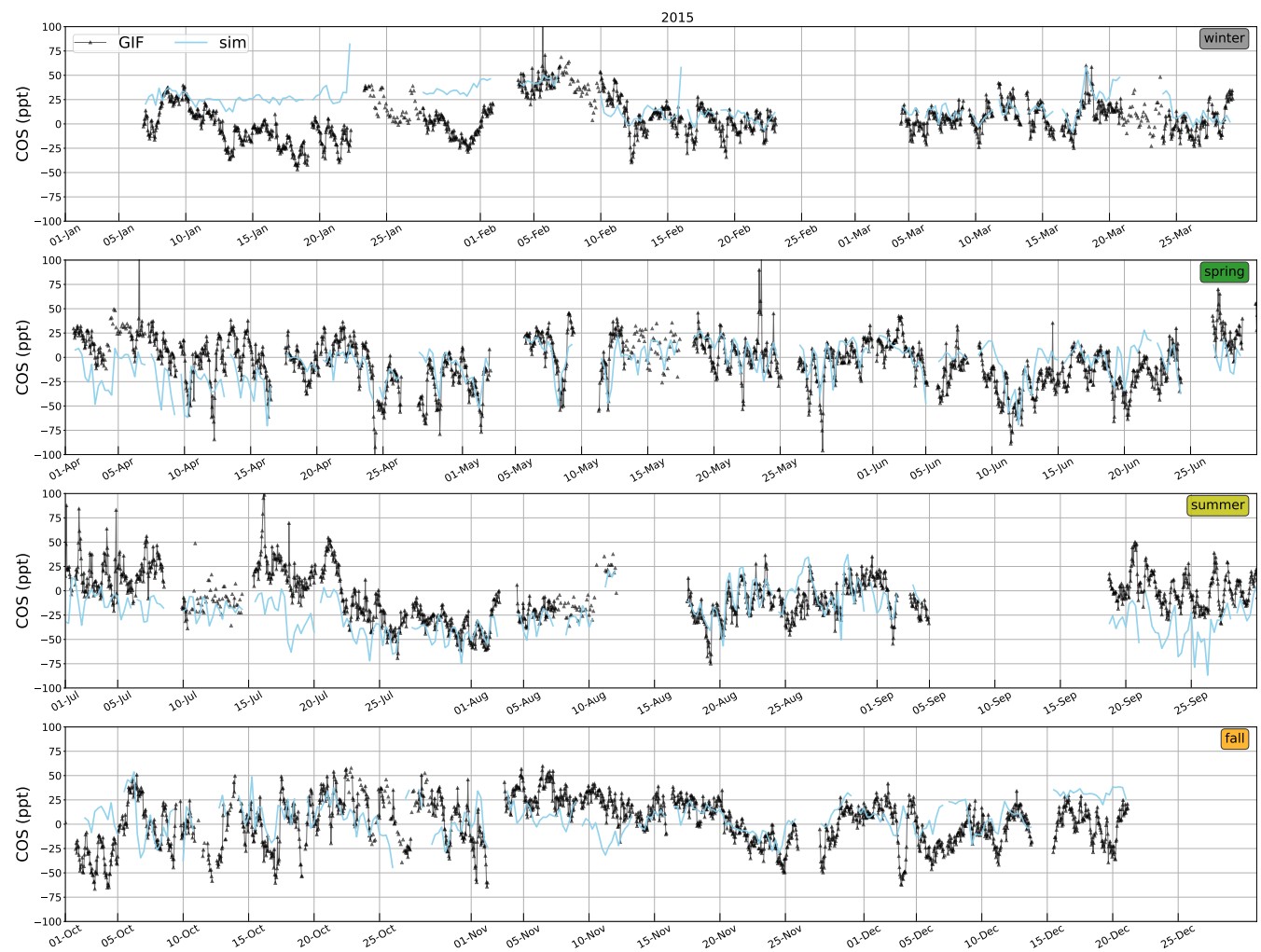

**Figure A2.** *Same as Fig. A1 for year 2015.*

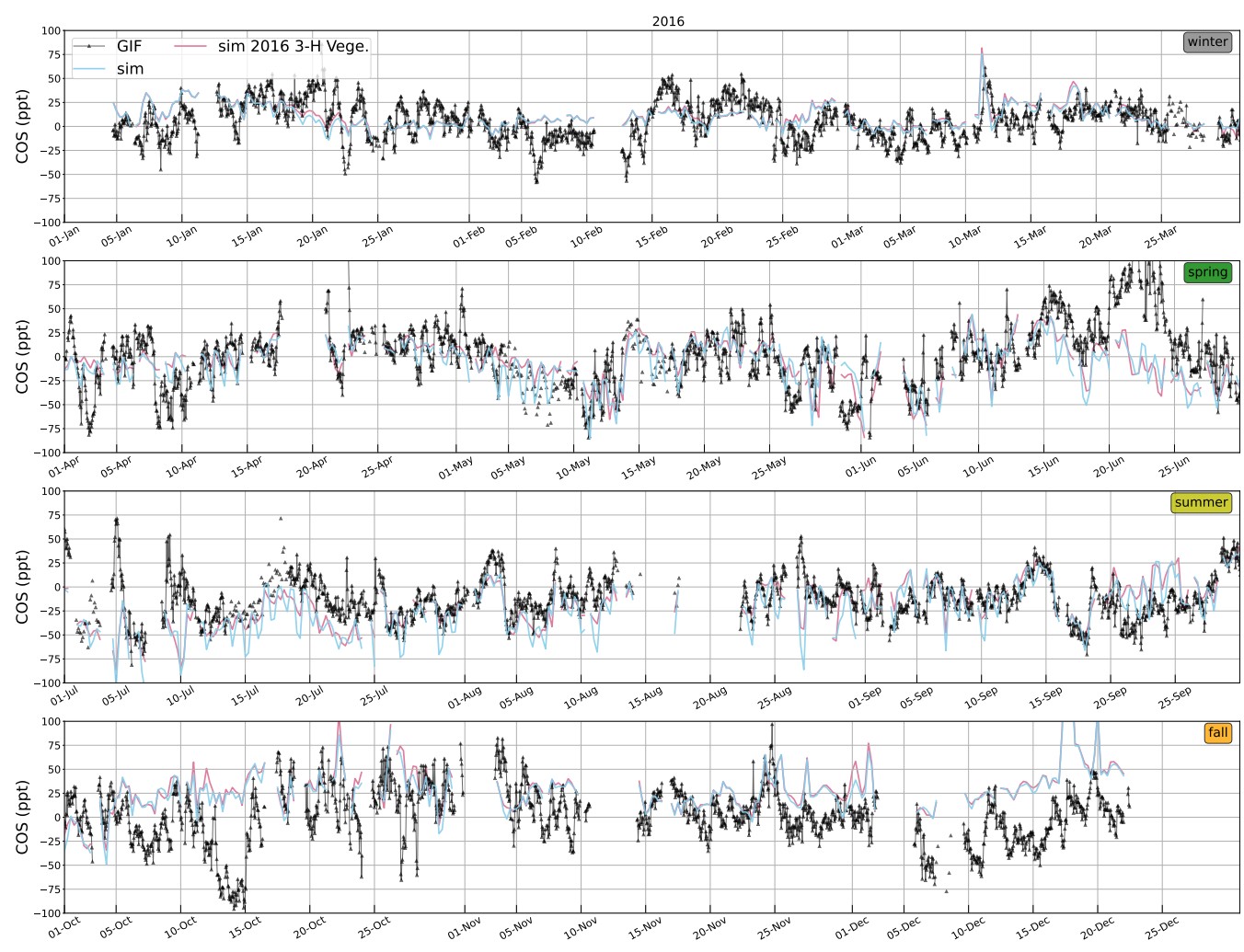

**Figure A3.** *Same as Fig. A1 for year 2016, with simulated contribution from the vegetation uptake with 3-hourly time-resolution (see Section 2.2 for details).*

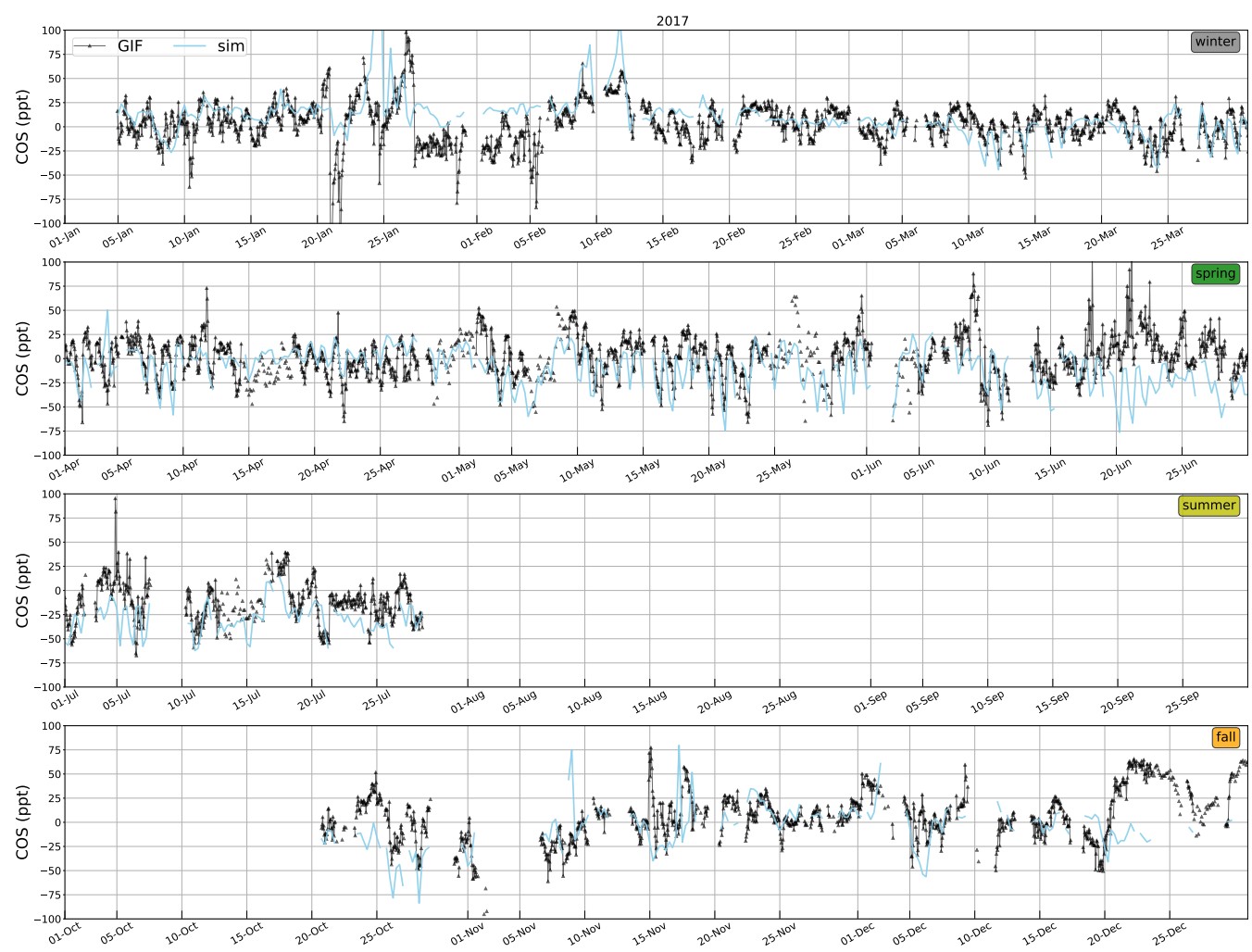

**Figure A4.** *Same as Fig. A1 for year 2017.*

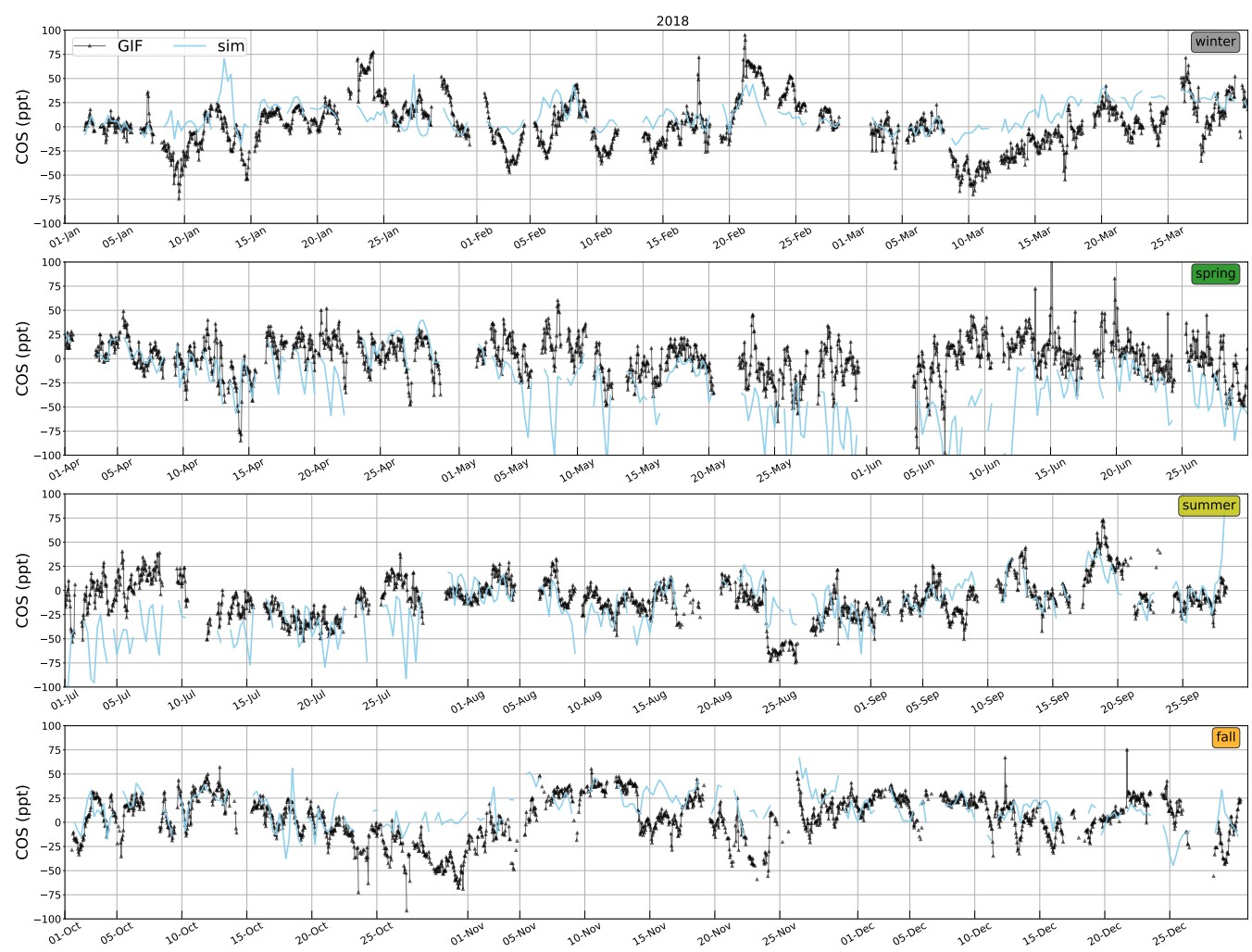

**Figure A5.** *Same as Fig. A1 for year 2018.*

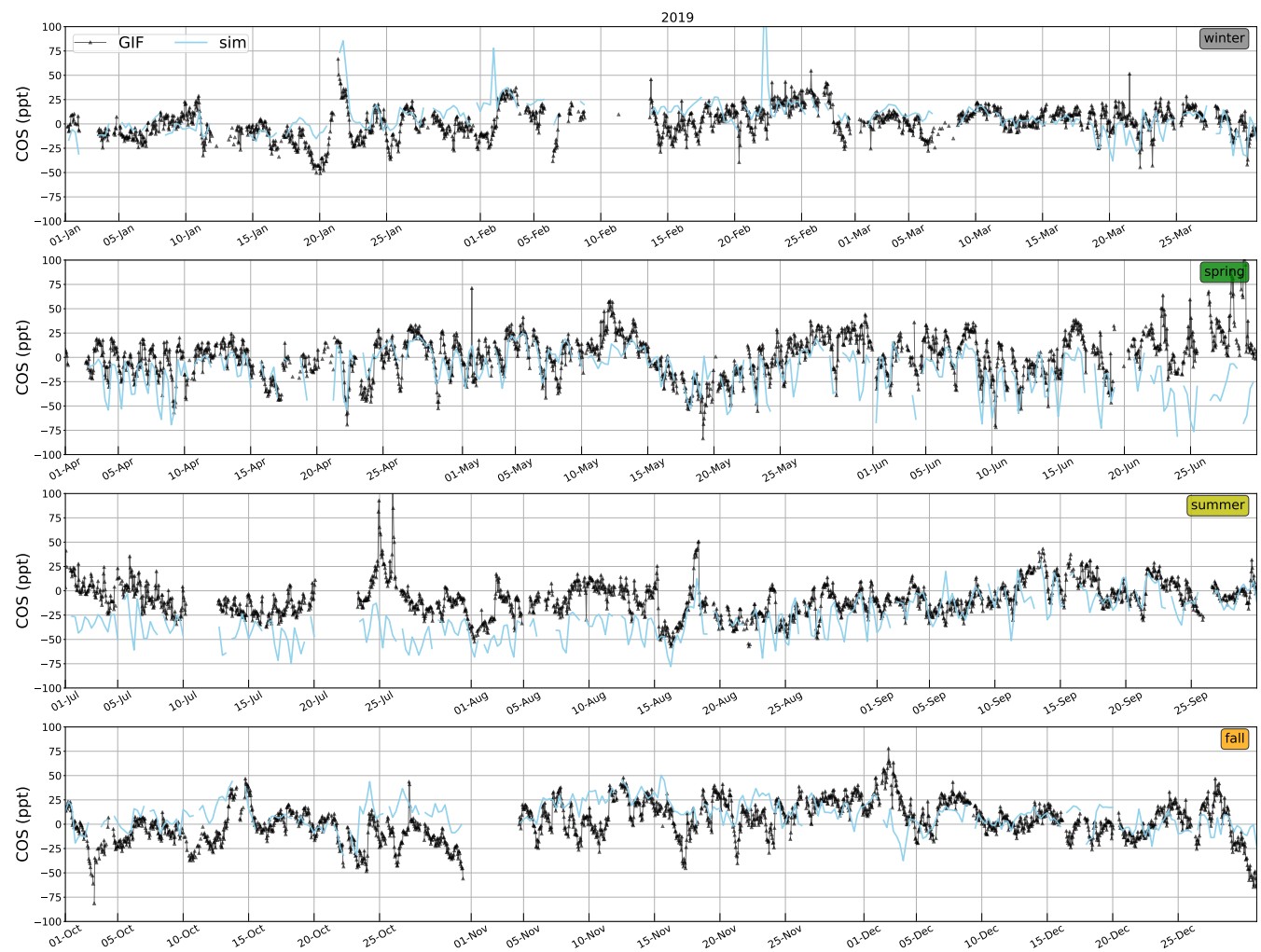

**Figure A6.** *Same as Fig. A1 for year 2019.*

## Appendix B: Contributions to COS mixing ratios at GIF

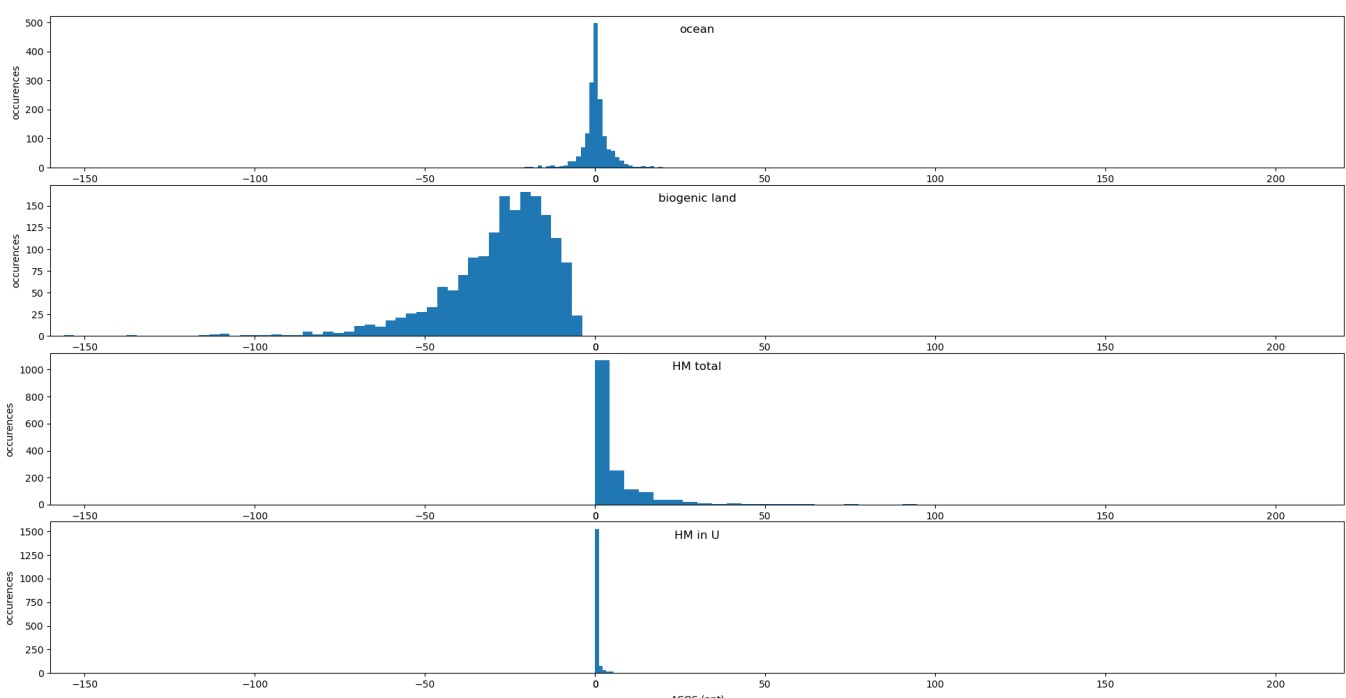

**Figure B1.** *Occurrences of contributions (ΔCOS, in ppt) to COS mixing ratios at GIF due to the background, the ocean, the vegetation and soil ("biogenic land") and our home-made anthropogenic emissions (total and only the Paris+Rouen area). Total number of occurrences: 1673 simulated values matching afternoon (12:00-18:00UTC) valid mean observations between 2014 and 2019.*

 **Appendix C: Statistical indicators of the performances of the model compared to the measurements at GIF for each contribution**

Table C1: *Statistical indicators of the performances of the model compared to the measurements at GIF for each contribution; indicators are either computed over the whole period, i.e., Aug. 2014 - Dec. 2019 (see Section 2.1 , "whole"), separated into four seasons ("winter" = January, February, March; "spring" = April, May, June; "summer" = July, August, September; "fall" = October, November, December) or only over the year 2016 for comparison with the available contribution by the 3-hourly varying vegetation. For each period, the means are during the afternoon (12:00-18:00 UTC, "pm"), the whole day ("day") or nighttime (21:00-02:00 UTC, "night"). Bias = mean difference model minus measurement; RMS = root mean square model minus measurements; corr = Pearson's correlation coefficient between model and measurement time series. HM indicates emissions from our home-made inventory (see Section 2.2.3 ).*

| Contribution | period | bias (ppt) | | | RMS (ppt) | | | corr | | |
|---|---|---|---|---|---|---|---|---|---|---|
| | | pm | day | night | pm | day | night | pm | day | night |
| Background | whole | 23 | 26 | 25 | 33 | 34 | 34 | 0.76 | 0.78 | 0.77 |
| | winter | 16 | 19 | 19 | 26 | 26 | 26 | 0.54 | 0.56 | 0.55 |
| | spring | 23 | 27 | 33 | 32 | 35 | 43 | 0.27 | 0.26 | 0.46 |
| | summer | 22 | 22 | 20 | 31 | 30 | 30 | 0.72 | 0.82 | 0.79 |
| | fall | 30 | 34 | 31 | 40 | 43 | 42 | 0.35 | 0.43 | 0.42 |
| Background + ocean | whole | 23 | 26 | 25 | 33 | 35 | 35 | 0.75 | 0.78 | 0.76 |
| | winter | 17 | 19 | 19 | 26 | 26 | 26 | 0.53 | 0.55 | 0.54 |
| | spring | 26 | 30 | 30 | 34 | 37 | 38 | 0.28 | 0.3 | 0.23 |
| | summer | 22 | 22 | 20 | 31 | 29 | 29 | 0.72 | 0.83 | 0.80 |
| | fall | 28 | 32 | 31 | 40 | 42 | 42 | 0.3 | 0.38 | 0.42 |
| Background + biogenic land | whole | -5 | -10 | -17 | 25 | 28 | 36 | 0.73 | 0.72 | 0.62 |
| | winter | -1 | 0 | -2 | 21 | 19 | 21 | 0.43 | 0.45 | 0.38 |
| | spring | -14 | -24 | -34 | 28 | 35 | 48 | 0.43 | 0.43 | 0.31 |
| | summer | -10 | -22 | -33 | 24 | 30 | 43 | 0.7 | 0.79 | 0.67 |
| | fall | 3 | 3 | 0 | 26 | 23 | 25 | 0.45 | 0.61 | 0.60 |
| Background + biogenic land | 2016 | -4 | -8 | -14 | 25 | 28 | 36 | 0.72 | 0.73 | 0.65 |
| Background + biogenic land with 2016 3-hourly vegetation | 2016 | -5 | -5 | -7 | 26 | 27 | 32 | 0.71 | 0.73 | 0.66 |
| Background + ocean + biogenic land | whole | -5 | -10 | -17 | 25 | 27 | 35 | 0.74 | 0.74 | 0.64 |
| | winter | -1 | 0 | -2 | 21 | 19 | 21 | 0.44 | 0.46 | 0.39 |
| | spring | -11 | -20 | -31 | 27 | 34 | 46 | 0.44 | 0.44 | 0.32 |

…/…

Table C1: *Statistical indicators of the performances of the model compared to the measurements at GIF for each contribution*
*(continued)*

| Contribution | period | bias (ppt) | | | RMS (ppt) | | | corr | | |
|---|---|---|---|---|---|---|---|---|---|---|
| | | pm | day | night | pm | day | night | pm | day | night |
| | summer | -10 | -22 | -33 | 25 | 30 | 43 | 0.71 | 0.81 | 0.70 |
| | fall | 1 | 1 | -3 | 27 | 24 | 26 | 0.41 | 0.57 | 0.57 |
| Background + ocean + biogenic land | 2016 | -3 | -7 | -13 | 25 | 27 | 36 | 0.73 | 0.75 | 0.65 |
| Background + ocean + biogenic land with 2016 3-hourly vegetation | 2016 | -4 | -4 | -7 | 26 | 26 | 32 | 0.72 | 0.75 | 0.68 |
| | whole | 100 | 165 | 231 | 151 | 230 | 358 | 0.11 | 0.02 | 0.02 |
| | winter | 103 | 144 | 176 | 164 | 214 | 292 | 0.1 | 0.19 | 0.17 |
| Background + anthro. Zumkehr | spring | 83 | 164 | 264 | 105 | 216 | 384 | -0.09 | -0.17 | -0.13 |
| | summer | 78 | 180 | 284 | 96 | 240 | 414 | 0.19 | -0.09 | -0.07 |
| | fall | 131 | 172 | 206 | 203 | 248 | 336 | 0.01 | -0.13 | -0.15 |
| | whole | 58 | 65 | 66 | 87 | 87 | 94 | 0.27 | 0.3 | 0.27 |
| | winter | 59 | 64 | 65 | 105 | 93 | 98 | 0.15 | 0.28 | 0.26 |
| Background + anthro. Zumkehr w/o Paris/Rouen | spring | 51 | 63 | 68 | 70 | 83 | 97 | -0.12 | -0.19 | -0.14 |
| | summer | 46 | 52 | 52 | 60 | 66 | 70 | 0.37 | 0.36 | 0.31 |
| | fall | 73 | 77 | 78 | 101 | 99 | 103 | 0.01 | -0.07 | -0.06 |
| | whole | 26 | 29 | 28 | 35 | 37 | 37 | 0.73 | 0.76 | 0.74 |
| | winter | 19 | 22 | 22 | 28 | 28 | 28 | 0.52 | 0.59 | 0.54 |
| Background + coal (HM) | spring | 25 | 29 | 29 | 34 | 37 | 38 | 0.21 | 0.19 | 0.15 |
| | summer | 24 | 24 | 22 | 32 | 32 | 32 | 0.71 | 0.8 | 0.77 |
| | fall | 34 | 37 | 36 | 44 | 46 | 46 | 0.29 | 0.36 | 0.39 |
| | whole | 25 | 28 | 27 | 34 | 36 | 36 | 0.74 | 0.77 | 0.75 |
| | winter | 18 | 21 | 21 | 27 | 28 | 27 | 0.51 | 0.55 | 0.55 |
| Background + viscose (HM) | spring | 24 | 29 | 29 | 33 | 36 | 38 | 0.23 | 0.2 | 0.15 |
| | summer | 23 | 24 | 22 | 32 | 32 | 32 | 0.71 | 0.81 | 0.78 |
| | fall | 32 | 36 | 36 | 43 | 45 | 45 | 0.32 | 0.38 | 0.41 |
| | whole | 27 | 31 | 30 | 37 | 39 | 39 | 0.71 | 0.74 | 0.72 |
| | winter | 21 | 24 | 24 | 30 | 30 | 30 | 0.47 | 0.55 | 0.51 |
| Background + coal (HM) + viscose (HM) | spring | 26 | 31 | 32 | 35 | 39 | 40 | 0.16 | 0.13 | 0.09 |
| | summer | 25 | 26 | 24 | 34 | 34 | 34 | 0.69 | 0.79 | 0.75 |

…/…

Table C1: *Statistical indicators of the performances of the model compared to the measurements at GIF for each contribution (continued)*

| Contribution | period | bias (ppt) | | | RMS (ppt) | | | corr | | |
|---|---|---|---|---|---|---|---|---|---|---|
| | | pm | day | night | pm | day | night | pm | day | night |
| | fall | 36 | 39 | 39 | 47 | 49 | 49 | 0.25 | 0.32 | 0.34 |
| Background + ocean + biogenic land + anthro. Zumkehr | whole | 72 | 129 | 190 | 129 | 198 | 323 | 0.14 | 0.05 | 0.02 |
| | winter | 85 | 125 | 155 | 149 | 196 | 274 | 0.11 | 0.21 | 0.17 |
| | spring | 49 | 117 | 206 | 71 | 172 | 335 | 0.05 | -0.11 | -0.10 |
| | summer | 46 | 136 | 231 | 67 | 203 | 372 | 0.26 | -0.07 | -0.06 |
| | fall | 102 | 138 | 170 | 179 | 219 | 307 | 0.03 | -0.1 | -0.13 |
| Background + ocean + biogenic land + anthro. Zumkehr w/o Paris/Rouen | whole | 29 | 29 | 24 | 66 | 59 | 66 | 0.33 | 0.39 | 0.32 |
| | winter | 41 | 44 | 44 | 93 | 77 | 82 | 0.16 | 0.31 | 0.28 |
| | spring | 17 | 16 | 10 | 39 | 44 | 60 | 0.11 | 0.04 | 0.00 |
| | summer | 13 | 8 | -1 | 34 | 34 | 41 | 0.49 | 0.53 | 0.41 |
| | fall | 43 | 44 | 42 | 76 | 70 | 73 | 0.05 | 0.04 | 0.06 |
| Background + ocean + biogenic land + coal HM + viscose HM | whole | -1 | -5 | -12 | 25 | 25 | 33 | 0.74 | 0.74 | 0.64 |
| | winter | 4 | 5 | 3 | 21 | 18 | 20 | 0.49 | 0.59 | 0.48 |
| | spring | -8 | -16 | -26 | 24 | 30 | 41 | 0.45 | 0.45 | 0.32 |
| | summer | -7 | -18 | -30 | 23 | 27 | 39 | 0.72 | 0.82 | 0.71 |
| | fall | 6 | 6 | 3 | 29 | 26 | 27 | 0.36 | 0.54 | 0.54 |