# Peer review of "Improved understanding of anthropogenic and biogenic carbonyl sulfide (COS) fluxes in Western Europe from long-term continuous mixing ratios measurements"

_EGUsphere, 2024_

## Referee Comment (RC1)

Review of Eguphere-2024-549

This is an interesting study to gain the knowledge about COS anthropogenic and biogenic emissions from the single atmospheric mixing ratio measurement site at GIF, France. The manuscript is well constructed and well written and the methodology to analyze the emission inventories and comparison with COS measurements is excellent. The study contributes new knowledge to COS measurement and modelling studies; therefore, I recommend publication after minor revision.

1. How is the measurement error, data quality and quality control of COS at GIF and comparison with COS measurement other locations? Please discuss them in the section of method, and/or in the discussion.
2. The analysis did not consider biomass burning emissions from COS. Why is that?
3. The authors compared two anthropogenic emission inventories. Is it possible to compare them on the same scale? For example, to compare Fig. 2a and Fig. 2c on the same unit, and estimate the total emissions at the same region.
4. DMS is also an important precursor of COS. Is there any industrial production of DMS that can explain some of the overestimated COS measurement at GIF?

Minor comments and technical corrections:
Page1, Line 3: "Moreover, COS atmospheric mixing ratio data are still too sparse to evaluate the estimations of these sources and sinks." While it is true that the COS mixing ratio data are sparse, but the evaluation of COS sources and sinks are always possible, e.g. using inverse model and satellite data assimilation.
Page 1, Line 4: "in the footprint a measurement site" to "in the footprint of a measurement site".
Page 1 Line 12: "ORCHIDEE" to "land surface model ORCHIDEE".
Page 1, Line 17: (Whelan et al., 2018, and references within), this I suggest to cite important papers explicitly.
Page 1 Line 17: "thinking" to "suggestions".
Page1, Line 20: here the authors introduced inverse method, but there are already a few studies of COS inversions. I recommend proper citations here.
Page 3 Line 6: "GgS/y" to "GgS.yr$^{-1}$" to keep consistency with other units. And elsewhere.
Page3 Line 17: "Remaud et al. (2023)" to "Remaud et al. (2023) and Ma et al. (2021)".
Page 3 Line 22: It is suggested to cite Montkza et al. (2007) here.
Page 6 Line 9: "horizontal resolution" to "horizontal resolution, respectively."
Page 6 Line 14: "GgS/y" to "GgS/yr". Please fix this unit throughout the text and keep them consistent.
Page 6 Line 21: "respectively)." to "respectively."
Page 6 Line 24: "1 degree ×1 degree" to "1°×1°", in order to keep consistent.
Page 6 Line 28: "The difference performance of our model with monthly and 3-hourly fluxes is evaluated in Sect. 3.1.". Maybe refer to "the different performances of our model with monthly and 3-hourly fluxes are evaluated in Sect. 3.1.".
Page 7 Line 4: "GgS", is it "GgS /yr" for the year 2012?
Page 7 Line 22: "in Tab. 2-3 of Belviso" to "in Table 2-3 of Belviso", and elsewhere.
Page 8 Line 11: "Barnes et al. (1994)". This paper discussed DMS conversion rate to COS about 0.7%. It is not about CS2 conversion to COS.

Page 8 Line 15: "So called" to "So-called".
Page 8 Line 16: "above ground." to "above ground level."
Page 8 Line 19: "(Tab. 1)" to "(Table 1)".
Page 10 Line 20: "coarse a variability" to "coarse variability".
Page 11 Table 2: "Background + coal (HM) + viscose (HM)" shows slightly larger bias and RMS, and smaller correlation than "Background + coal (HM)" or "Background + viscose (HM)". Does this indicate the sum of coal and viscose from HM inventory is less accurate than single coal or viscose?
Page 16 Line 2: "Zanchetta et al., 2023" to "Zanchetta et al., (2023)".
Page 16 Line 12: "Ramonet et al., 2011" to "Ramonet et al., (2011)".
Page 16 Line 16: "(Remaud et al., 2023)" to "Remaud et al., (2023)".

---

## Author Comment (AC1)

**Author responses to reviews to EGUSPHERE-2024-549 "Can we gain knowledge on COS anthropogenic and biogenic emissions from a single atmospheric mixing ratios measurement site?"**

Antoine Berchet[1, *], Isabelle Pison[1, *], Camille Huselstein[1], Clément Narbaud[1], Marine Remaud[1, 2], Sauveur Belviso[1], Camille Abadie[1] and Fabienne Maignan[1]

[1]Laboratoire des Sciences du Climat et de l'Environnement, CEA-CNRS-UVSQ, Gif-sur-Yvette, France
[2]now at: Faculty of Science, A-LIFE, Vrije Universiteit Amsterdam, 1081 HV Amsterdam, the Netherlands
[*]Correspondance: antoine.berchet@lsce.ipsl.fr, isabelle.pison@lsce.ipsl.fr

We thank the referees for their detailed and fruitful comments on our manuscript. We reproduce below their reviews and embed our responses in bold blue text in their comments. Revised section of the manuscript are reproduced in italic blue. We also provide a track-change manuscript at the end of the present document.

**1 Referee #1: Mary Whelan**

This paper is a good application of the lead author's work making FLEXPART more accessible. The treatment of the CS2 fields here is an improvement for the field, transforming emissions of CS2 to atmospheric COS with a reasonable conversion rate.

I have a few suggestions for making this work more impactful.

**1.1 Main comments**

**1.1.1 Zumkehr inventory**

It is unclear why the Zumkehr inventory was used without the improvements suggested by Belviso et al., (2023). Belviso et al., demonstrated the drawbacks to this inventory fairly conclusively! It might make sense to make the necessary changes and move on. Additionally, naming the improved anthropogenic scheme "home-made" is confusing. Zumkehr was also likely working from home. Using the designation "This Study" is clearer.

**Belviso et al. (2023) assessed discrepancies between Zumkehr's inventory and measurements at a rather local scale. The objective of the present study is to assess COS fluxes at a regional scale. Thus we chose to re-use Zumkehr as such as a reference inventory, and then we propose an updated inventory based on industrial declarations at the European scale as a new starting point, leading to significant and convincing improvements. We clarify our objectives and rationals in the new manuscript.**

We revise the last paragraph of our introduction as follows:

20 *Therefore, the present study aims at building a set-up that makes it possible to quantitatively assess the anthropogenic and biogenic COS fluxes at the regional scale, i.e., in the footprint of one measurement site in Western Europe at a seasonal to diurnal time resolution over a period of half a decade. For this, we use the continuous time series of COS mixing ratios measured in the Paris area from summer 2014 to the end of year 2019, as described in obsgif. We compare them to the concentrations simulated from marine, biogenic and anthropogenic fluxes in the area of interest (detailed in flux) combined*

25 *to the contribution due to the rest of the world (bckg) by the modeling tool described in model. After an assessment of the general performances of the model (general), we are able to quantitatively evaluate the anthropogenic sources from Western Europe as estimated by Zumkehr et al. (2018) (hereafter referred to as "Zumkehr's inventory") and by our more targeted inventory (anthro), confirming discrepancies from Belviso et al. (2023) in Zumkehr's inventory in France in particular, but also in Western Europe in general. Contrary to Belviso et al. (2023), the present study goes one step further by quantitatively*

30 *assessing discrepancies in Zumkehr's inventory and by proposing a new inventory based on industrial emission declaration in the European Union. Having more reliable anthropogenic emissions, we can inquire into biogenic emissions, which is one of the main original purpose of studying COS. We study the seasonal and diurnal cycles of biogenic fluxes, based on the ORCHIDEE and SiB4 processed-based land surface models (vegesink); this allows us to point to strengths and weaknesses in the two models.*

35 ### 1.1.2 Land uptake

To improve our understanding of the atmospheric OCS balance, it would make sense to use a different biogenic uptake dataset instead, such as SiB4. Comparing SiB4 to ORCHIDEE in this framework might yield some interesting insights into our treatment of plant uptake.

 **We recompute our simulations based on SIB4 in complement to ORCHIDEE. We thank the reviewer for this very**

40 **interesting suggestion that we integrate in the updated manuscript. We updated Sect. 2.2.2 accordingly, adding description of the SiB4 model.**

 **We include results from SiB4 in Fig. 3 and 4 and in Sect. 3.3., as well as in the conclusion.**

 **Overall, SiB4 seems to better reproduce the diurnal cycle of fluxes, especially in Spring, compared to ORCHIDEE. The magnitude of Summer fluxes leads to better agreement with observations when using ORCHIDEE than SiB4.**

45 ### 1.1.3 Temporal coverage of footprints

10, 15-23 and Section 3.3, Figure 4: The footprints used here are calculated throughout the day. In efforts involving WRF-STILT, one recommended approach is to average afternoon tower measurements when the boundary layer is typically well-mixed. As the authors note, it could be that there is variation in the tower concentrations because of shifts in the mixing layer rather than shifts in the plant uptake or ocean background. While OCS is undoubtedly being taken up at night by partially-closed

50 stomata, there is also a dynamic shift in the state of the mixed layer in contact with the tower. How well does FLEXPART treat the boundary layer? Perhaps some explanation needs to be included here on the uncertainty introduced by taking footprints at

different times of day. Alternatively, averaging concentrations when the layer is stable would side-step some of the synoptic-scale complications.

**We agree that FLEXPART (and any transport model) is most suitably compared with observations during afternoon time, when the boundary layer is well mixed. This is why general performances and assessment of synoptic scale transport in regards to anthropogenic emissions is done using afternoon data only. This is clarified in the update manuscript.**

**As there is basically no non-biogenic OCS fluxes in the vicinity of the observation site GIF, we choose to use the diurnal cycle of observations and simulations to deduce variations in the magnitude and sign of local OCS biogenic fluxes. Such an approach relates to a simulation based Radon-method.**

**Although we are confident in FLEXPART's capability in reproducing well mixed afternoon conditions, there are indeed uncertainties in nighttime simulations, in particular related to the parametrization of PBL mixng, as well as in the value of the PBL height, as provided by ECMWF ERA-5 data at night. Such uncertainties are hard to quantify, but they would only impact the values of our simulations, and not our overall conclusions that stomatae behaviour is not well reproduced in ORCHIDEE and should be further improved. We clarify these points in the updated manuscript in the conclusion and in Sect. 3.3.**

**1.2 Minor comments**

1. The title of the article needs to be revised. "Can we gain knowledge on COS anthropogenic and biogenic emissions from a single atmospheric mixing ratios measurement site?" is a yes or no question. Here we learn that OCS uptake is underestimated at night and that there are limitations in the anthropogenic inventory. Perhaps something like, "improvements to anthropogenic and biogenic fluxes of OCS in Western Europe".

   **Thank you for this comment. We update the title to "Improved understanding of anthropogenic and biogenic carbonyl sulfide (COS) fluxes in Western Europe from long-term continuous mixing ratios measurements".**

2. Abstract: use of the word "fluxes" versus "emissions" leads to confusing sentence constructions. Page 1, line 3: "Moreover, COS atmospheric mixing ratio data are still too sparse to evaluate the estimations of these sources and sinks." - do you mean on the global scale?

   **We clarify the use of emissions/fluxes in the abstract; we rather mention uptake/source. Data are too sparse at the global scale already, despite attempts to do global inversions, but the issue is even more striking for Europe.**

3. 4, 25: I don't see the Mace Head data plotted on Figure 1. Is something missing in the pdf?

   **NOAA MHD data are depicted as red dots in Fig. 1. We will double check on the final version of the manuscript that these points appear properly.**

4. Figure 1. Categories in the legend are missing in the graph in (a) and (b). For (a), it appears that the brown is background and the black dots, which end in 2015, are observations from GIF?

**There might be a display bug in some computers. The categories appear on our system. We will double check this on different system during proof-reading.**

5. 6, 5-6: There are good reasons to leave out these fluxes. Making the assumptions explicit, e.g. these fluxes do not contribute meaningfully to the observations, would be useful here.

**These fluxes are very small in Europe compared to anthropogenic, biogenic and ocean fluxes. They rather play a role at the global scale. We clarify and state explicitly these reasons in the new manuscript.**

6. 6, 13: similar results on the global scale?

**That is correct. we clarify this point**

7. 6, 27: COS uptake is related to stomatal conductance. For C3 and C4 plants, we expect stomatal conductance to increase during the day. Stomatal conductance never quite gets to zero when closed at night or in drought. If the ORCHIDEE plant uptake is based on Berry et al 2013, uptake should vary with stomatal conductance in the model as well as in reality.

**The mentioned sentence is indeed misleading. We replace it by:**

**"In particular, biogenic fluxes exhibit a significant diurnal cycle. Indeed, vegetation COS uptake is regulated by stomatal conductance. There is a residual uptake during nighttime due to incomplete stomatal closure, and a stronger uptake during daytime when stomatal conductance increases."**

8. 7, 7-9: It is unclear what this means. Are the national emissions erroneously attributed to urban hot spots? And we know that these hot spots are an artifact? But looking at Figure 2, it appears that Zumkehr spreads national emissions over the entire country, whereas the improved inventory locates emissions in a collection of hotspots.

**Zumkehr, when using generic industry proxy, spreads OCS anthropogenic fluxes over the whole country with hotspots around large megalopolises. In our inventory, OCS industrial emissions are indeed limited to a handful of hotspots (with only one coal-power power plant and very few industrial sites, not located near the biggest megalopolises), but not correlated to large urban centers or other proxies. We clarify this sentence.**

9. Figure 2. It would be good to have a color bar or similar for the contours in 2(f) so that the figure can be understood without digging into the text.

**We agree that more stand-alone information about the contour in the figure is needed. A colorbar is not as informative as expected, regarding the arbitrary units of the sensitivity. We choose to clarify the sub-title for figure 2f**

**1.3 Concluding remarks**

Thanks for the work in interpreting these data. It would be excellent to use this work to initiate a COS observing network analogous to ICOS.

Sincerely,

Mary Whelan

Citation: https://doi.org/10.5194/egusphere-2024-549-RC2

**We thank Mary Whelan for her fruitful comments on our manuscript.**

**2 Referee #2**

**2.1 Main remarks**

This is an interesting study to gain the knowledge about COS anthropogenic and biogenic emissions from the single atmospheric mixing ratio measurement site at GIF, France. The manuscript is well constructed and well written and the methodology to analyze the emission inventories and comparison with COS measurements is excellent. The study contributes new knowledge to COS measurement and modelling studies; therefore, I recommend publication after minor revision.

**Thank you for your fruitful comment. We address them below.**

1. How is the measurement error, data quality and quality control of COS at GIF and comparison with COS measurement other locations? Please discuss them in the section of method, and/or in the discussion.

**The description of observations and quality control was already detailed in previous publications and we did not want to add too much details in the present study. Still, we agree that we overlooked some key information and we update the manuscript accordingly.**

2. The analysis did not consider biomass burning emissions from COS. Why is that?

**Biomass burning in Europe, in particular in the vicinity of GIF in France, play a very limited role in OCS concentrations. They would play a role at the global scale, but global scale concentrations from the background already account for biomass burning and long range transport. We clarify this point in the updated manuscript.**

3. The authors compared two anthropogenic emission inventories. Is it possible to compare them on the same scale? For example, to compare Fig. 2a and Fig. 2c on the same unit, and estimate the total emissions at the same region.

   **It is not possible to use the same scale in Fig 2a and 2c as 2a is built on area sources, whereas 2c is an ensemble of point sources. The total emissions at the European scale are roughly a factor of 6 superior in Zumkehr, compared to our inventory, explaining the large discrepancies in simulations. We highlight this point in the corresponding section describing inventories**

4. DMS is also an important precursor of COS. Is there any industrial production of DMS that can explain some of the overestimated COS measurement at GIF?

   **To our knowledge DMS rather contributes to COS through oceanic emissions and later oxidation. Oceanic emissions have a very smooth impact on inland GIF concentrations. Thus we do not expect missed observed peaks to be explained by DMS fluxes. We still include some elements on DMS in the updated manuscript.**

**2.2 Minor comments and technical corrections**

1. Page1, Line 3: "Moreover, COS atmospheric mixing ratio data are still too sparse to evaluate the estimations of these sources and sinks." While it is true that the COS mixing ratio data are sparse, but the evaluation of COS sources and sinks are always possible, e.g. using inverse model and satellite data assimilation.

   **We agree that our statement is misleading. Observations are very sparse and allow for only rather coarse estimation of fluxes, with regional estimation very challenging.**

2. Page 1, Line 4: "in the footprint a measurement site" to "in the footprint of a measurement site".

   **We thank the reviewer for this typo**

3. Page 1 Line 12: "ORCHIDEE" to "land surface model ORCHIDEE".

   **We thank the reviewer.**

4. Page 1, Line 17: (Whelan et al., 2018, and references within), this I suggest to cite important papers explicitly.

   **We include most important references from Whelan.**

5. Page 1 Line 17: "thinking" to "suggestions".

   **We agree on the better wording**

6. Page 1, Line 20: here the authors introduced inverse method, but there are already a few studies of COS inversions. I recommend proper citations here.

**We agree that numerous studies, bottom up and top down have been made since the years 2000, with increased interest in the last 5 years. We include more publications, even though it is hard to include all of them. We are sorry if we overlooked some key publications.**

7. Page 3 Line 6: "GgS/y" to "GgS.yr-1" to keep consistency with other units. And elsewhere.

**We modified units for consistency**

8. Page3 Line 17: "Remaud et al. (2023)" to "Remaud et al. (2023) and Ma et al. (2021)".

**We are sorry for not including this citation here, even though it was included elsewhere.**

9. Page 3 Line 22: It is suggested to cite Montkza et al. (2007) here.

**Thank you for this suggestion**

10. Page 6 Line 9: "horizontal resolution" to "horizontal resolution, respectively."

**Thank you for this suggestion**

11. Page 6 Line 14: "GgS/y" to "GgS/yr". Please fix this unit throughout the text and keep them consistent.

**The notation is made consistent throughout the whole manuscript now**

12. Page 6 Line 21: "respectively)." to "respectively."

**This paragraph has been updated with description of SiB4 and this typo has been fixed as well**

13. Page 6 Line 24: "1 degree ×1 degree" to "1o×1o", in order to keep consistent.

**Thank you for this correction.**

14. Page 6 Line 28: "The difference performance of our model with monthly and 3-hourly fluxes is evaluated in Sect. 3.1.". Maybe refer to "the different performances of our model with monthly and 3-hourly fluxes are evaluated in Sect. 3.1.".

**Thank you for this correction.**

15. Page 7 Line 4: "GgS", is it "GgS /yr" for the year 2012?

**It is indeed per year. It is clarified now.**

16. Page 7 Line 22: "in Tab. 2-3 of Belviso" to "in Table 2-3 of Belviso", and elsewhere.

**We changed for the proposed notation.**

17. Page 8 Line 11: "Barnes et al. (1994)". This paper discussed DMS conversion rate to COS about 0.7%. It is not about CS2 conversion to COS.

    **We agree on the reviewer that Barnes et al. was not well used by Zumkehr et al. We still keep the 87% conversion rate from Zumkehr.**

18. Page 8 Line 15: "So called" to "So-called".

    **Thank you for this correction.**

19. Page 8 Line 16: "above ground." to "above ground level."

    **Thank you for this correction.**

20. Page 8 Line 19: "(Tab. 1)" to "(Table 1)".

    **Thank you for this correction.**

21. Page 10 Line 20: "coarse a variability" to "coarse variability".

    **Thank you for this correction.**

22. Page 11 Table 2: "Background + coal (HM) + viscose (HM)" shows slightly larger bias and RMS, and smaller correlation than "Background + coal (HM)" or "Background + viscose (HM)". Does this indicate the sum of coal and viscose from HM inventory is less accurate than single coal or viscose?

    **This table only analyses long term variability. Coal and viscose emissions have impact only as peaks, which are most of the time the same between coal and viscose. With only one site, we cannot dig in systematically on peaks, as their magnitude is difficult to reproduce. The take-home message is that with our inventory, we obtain correct magnitude. We clarify this message.**

23. Page 16 Line 2: "Zanchetta et al., 2023" to "Zanchetta et al., (2023)".

    **This reference is already in brackets, hence the year left-out of brackets**

24. Page 16 Line 12: "Ramonet et al., 2011" to "Ramonet et al., (2011)".

    **Same as above**

25. Page 16 Line 16: "(Remaud et al., 2023)" to "Remaud et al., (2023)".

    **Thank you for this correction.**

**References**

Belviso, S., Pison, I., Petit, J.-E., Berchet, A., Remaud, M., Simon, L., Ramonet, M., Delmotte, M., Kazan, V., Yver-Kwok, C., and Lopez, M.: The Z-2018 emissions inventory of COS in Europe: A semiquantitative multi-data-streams evaluation, Atmospheric Environment, 300, 119 689, https://doi.org/10.1016/j.atmosenv.2023.119689, 2023.

225 Zumkehr, A., Hilton, T. W., Whelan, M., Smith, S., Kuai, L., Worden, J., and Campbell, J. E.: Global gridded anthropogenic emissions inventory of carbonyl sulfide, Atmospheric Environment, 183, 11–19, https://doi.org/10.1016/j.atmosenv.2018.03.063, 2018.

---

## Author Response (AR2)

**Author responses to reviews to EGUSPHERE-2024-549 "Improved understanding of anthropogenic and biogenic carbonyl sulfide (COS) fluxes in Western Europe from long-term continuous mixing ratios measurements"**

Antoine Berchet[1, *], Isabelle Pison[1, *], Camille Huselstein[1], Clément Narbaud[1], Marine Remaud[1, 2], Sauveur Belviso[1], Camille Abadie[1] and Fabienne Maignan[1]

[1]Laboratoire des Sciences du Climat et de l'Environnement, CEA-CNRS-UVSQ, Gif-sur-Yvette, France
[2]now at: Faculty of Science, A-LIFE, Vrije Universiteit Amsterdam, 1081 HV Amsterdam, the Netherlands
[*]Correspondance: antoine.berchet@lsce.ipsl.fr, isabelle.pison@lsce.ipsl.fr

**We thank the referees for their detailed and fruitful comments on our manuscript. We reproduce below their reviews and embed our responses in bold blue text in their comments. Revised sections of the manuscript are reproduced in italic blue.**

**We also provide a track-change manuscript.**

**1 Report #1 by referee #3: Wu Sun**

In this manuscript, Berchet et al. use a five-year record of atmospheric COS concentrations measured at a Paris suburb site to examine anthropogenic and biogenic COS fluxes over Western Europe. Given the sparsity of COS observations globally, it is encouraging to see this work leveraging long-term COS observations at a single site to shed light on regional COS flux components, aided by a Lagrangian transport model. The study shows promise in enhancing the usability of COS as a photosynthetic tracer at regional scales.

While the revised manuscript has been improved in response to prior review comments, I find some remaining issues. I have a few suggestions for improving the robustness and clarity of the findings.

**We thank Wu Sun for these comments on our manuscript and we hope we answered all of them in the following.**

**1.1 Main comments**

To alleviate potential bias from nighttime boundary layer parameterization, I suggest running additional tests to re-examine the diagnostics for different flux combinations (Table 2) using only mid-afternoon COS observations. Parameterization of the nighttime boundary layer is notoriously challenging and can lead to transport errors (e.g., Díaz-Isaac et al., 2018; Lopez-Coto et al., 2020; Monteiro et al., 2024). As a result, most Lagrangian inverse modeling studies assimilate only observations during well-mixed afternoon conditions, which both the authors and the reviewer, Dr. Whelan, have acknowledged. Lacking information on how well FLEXPART represents the nighttime boundary layer, I am not confident that bias in simulated nighttime

concentrations could be attributed solely to the representation of nighttime stomatal conductance in terrestrial biosphere models (ORCHIDEE and SiB4). While I agree with some of the reasoning in the authors' response, I do not see this issue fully addressed. I reckon that the simplest way to test this issue could be to redo a set of analyses using only mid-afternoon COS observations and see if error diagnostics still follow the patterns in Table 2. As the atmosphere integrates fluxes day and night, afternoon observations should still be sensitive to fluxes in previous nights within the footprint area.

**We think there is a misunderstanding: in the Table 2, we actually use observations during the afternoon i.e. 12:00-18:00 UTC, which corresponds in France to 1 p.m.-7 p.m. local time in winter and 2 p.m.-8 p.m. in summer. We have clarified this in the legend of Table 2 - now Table 3:** *Statistical indicators of the performances of the model compared to the measurements at GIF for each contribution, based on the daily afternoon (12:00-18:00 UTC) means of simulated and measured mixing ratios; [. . . ]* **and have added a general explanation in the text at the beginning of Section 3.1:** *In the following, we use the afternoon (12:00-18:00 UTC) daily averages of measured and simulated mixing ratios because the model is assumed to better represent the vertical mixing in the afternoon so that the comparison to measurements highlights the discrepancies in fluxes compared to the model's errors.* **See also the answer to the next comment.**

On the evaluation of anthropogenic COS flux estimates, because point sources of viscose and coal industries are largely collocated (Fig. 2c), their influences on observed COS concentrations would be highly correlated. Therefore, it seems challenging to evaluate COS fluxes from these two sectors separately. On the other hand, because changes in the wind direction may allow the site to pick up flux signals coming from different regions (Paris/Rouen, Benelux, British Isles, etc.), it would seem more helpful to focus on contributions from different regions rather than different industries that are collocated in space.

**We agree that if both sectors were co-located, discriminating their contributions would be impossible. Nevertheless, they are actually not located so close to each other that the station at GIF cannot discriminate. For example, in France, where there is almost no coal industries but three coal power plants on the coasts in the Channel, on the Atlantic ocean and on the Mediterranean sea, the viscose industry is found in areas remote from the coasts i.e. in the North and in the East (see Fig. 2c in the paper and the zoom for France in Figure R1 here). The two activities also differ in their time profile e.g. coal COS emissions have an Inter-Annual Variability (IAV) derived from the IAV of $CO_2$ emissions due to power plants whereas viscose industry has constant emissions. This is why we chose to assess the emissions both for sectors and for different areas (see Fig. 2d for the areas definitions).**

[Figure]

[Figure]

**Figure R 1.** *Zoom of Fig. 2c over France. Triangle: viscose industry; circle: non-viscose industry i.e. here, coal industry, mainly power plants; amount of emissions shown by diameter and color, according to the scale on the left hand-size. Star: location of GIF station.*

On the evaluation of biogenic COS fluxes, given that the focus is on nighttime bias, presumably resulting from underestimated nighttime stomatal conductance, and on summertime bias caused by potential emissions from certain crops, it may help to further assess bias, RMSE, and correlation (following Table 2) for daytime and nighttime measurements and by season. Model bias in representing nighttime COS uptake has been known since Kooijmans et al. (2017), and it would be better to give quantitative insights beyond reiterating this message. I suggest some simple tests along this line, for example, treating nighttime and daytime ecosystem fluxes as two separate tracers and checking how much nighttime fluxes need to be bumped up in order to match observed COS concentrations. This would provide more detailed insights into model bias and help inform relevant process representations in terrestrial biosphere models.

**We agree that the reader may wish to see the whole set of statistics, following Table 2. We added supplementary material in Appendix C, here in Table R1, and refer to it in the main text:** *In the following, we use the afternoon (12:00-18:00 UTC) daily averages of measured and simulated mixing ratios because the model is assumed to better represent the vertical mixing in the afternoon so that the comparison to measurements highlights the discrepancies in fluxes compared to the model's errors. More detailed results for the whole day or nighttime and per season are shown in Table C1.*

**The idea of using two tracers for nighttime and daytime ecosystem fluxes and to evaluate how much nighttime fluxes must be changed to match the observations may be applied in atmospheric inversion to species such as $CO_2$ or $CH_4$ in Europe, where a whole network of measurement sites (e.g., ICOS) is available. With one station only for COS, we lack information for estimating the absolute values of the fluxes: only differences can be quantified. We added a sentence in Section 3.3 to clarify this point:** *Disentangling the main contribution of mismatches between observations and simulations is very challenging with one site only. Differences at the synoptic scale, as represented in Figure 1, can originate from erroneous background, transport discrepancies, or from incorrect fluxes. Moreover, with one site only, absolute values of the biogenic fluxes cannot be systematically estimated. We therefore focus on the variations through time, particularly between day and night and seasonally. As represented in the time series in Appendix A, fine scale temporal variability, corresponding to the diurnal cycle, is well reproduced during some periods of the year, especially in Spring. The diurnal cycle is dominated by local*

[revised manuscript text omitted]

Lastly, the study could give readers a high-level view of whether anthropogenic or biogenic COS fluxes dominate the variability in observed COS drawdown or enhancement (ΔCOS). Comparing Fig. 1d and Fig. 4, it appears the study domain is dominated by biogenic COS fluxes, but I'm not sure. I suspect that many in the carbon cycle research community would care about this question, but I don't see it explicitly answered.

**We agree that our text was not explicit enough. We changed the title of Section 3.1 "General performances of the model at GIF",** which is now *General patterns of simulated and measured COS concentrations at GIF* **and added a few sentences to clarify the relative weights of the contributions** *At GIF, the seasonal variations of COS mixing ratios are dominated by the contributions of the background and ocean i.e. by large scale fluxes; the variations at shorter time scales (week or day) are driven by the biogenic land contribution (Belviso et al., 2023). Finally, depending on the wind speed and direction, anthropogenic emissions may dominate for short episodes of high concentrations (see section 3.3.2 Selected winter*

*episodes in Belviso et al., 2023).* **as well as in the abstract** *At GIF, the seasonal variations of COS mixing ratios are dominated by the contributions of the background and ocean; the weekly to daily variations are driven by the biogenic land contribution; anthropogenic emissions may dominate for short episodes of high concentrations.* **and in the general conclusion** *At GIF, the*
85 *seasonal variations of COS mixing ratios are dominated by the contributions of the background and ocean; the weekly to daily variations are driven by the biogenic land contribution; anthropogenic emissions may dominate for short episodes of high concentrations (Belviso et al., 2023)..*

**1.2 Specific comments**

1. L4: "at a better scale than the global scale" –> "at the regional scale"

90 **OK, changed to** *Moreover, COS atmospheric mixing ratio data are still too sparse to evaluate the estimations of these sources and sinks at the regional scale.*

L13–14: "We find that the net ecosystem COS uptake simulated by both ORCHIDEE and SiB4 is underestimated in winter at night" - As I suggested above, it would be better to redo the evaluation with mid-afternoon COS observations only to confirm if the bias results from nighttime stomatal conductance representation.

95 **See our answer to the third main comment and the new supplementary Table in Appendix C. The statistical indicators computed with the background plus the ocean plus the biogenic land contribution (Table R1, lines "Background + ocean + biogenic land") confirm that the diurnal cycle of the fluxes plays a role in the poor nighttime agreement between the simulations and the measurements. The vegetation uptake is only a small portion of the terrestrial uptake (see Maignan et al. 2021): too weak a soil uptake could explain part of the discrepancy. The**
100 **remainder of the discrepancies during the night is indeed possibly due the transport. The underestimation of the ecosystem COS uptake by ORCHIDEE and SiB4 in winter at night is next to be precisely quantified, which is not in the scope of this work using only one station.**

L16–17: "In Summer, both models properly represents fluxes, with better agreement from ORCHIDEE in terms of magnitudes." - What about potential COS emission episodes?

105 **As stated in the main text l.79-80: "Some natural processes emitting COS in the atmosphere are still only suspected, for example in plants used in agriculture (Belviso et al., 2022a; Maseyk et al., 2014; Bloem et al., 2012). Therefore, such emissions are as yet no taken into account in ORCHIDEE, SiB4 nor any other model to our knowledge. Nevertheless, these suspected COS emissions by crops (rapeseed in this case) are very localised in time (at the end of the summer) and are not expected to affect the seasonality. In addition to these possible COS**
110 **emissions due to crops, soils may emit COS, even though they are overall a net sink of COS. These emissions are taken into account in ORCHIDEE and SiB4 (each model with its own parameterizations) and high-temperature situations may make soils a net source of COS. We modified in the summary** *In Summer, both models represent fluxes sufficiently well, with better agreement from ORCHIDEE in terms of magnitudes.*

L24–37: Since this study is not an atmospheric inversion, I don't see a need to introduce it. Consider focusing on the forward model application of linking flux fields to mixing ratios.

**We agree that this study is not an atmospheric inversion. Nevertheless, we feel that it is important to keep this study connected to the community of atmospheric inversion. COS has been studied by the $CO_2$ flux community for estimating $CO_2$ fluxes linked to photosynthesis by atmospheric inversion. We think it is important to link this work to this community because it may help in assessing the feasibility of using COS measurements for constraining $CO_2$ fluxes. This is particularly relevant for network planning and funding.**

L40–80: This part feels overly detailed as an introduction. I suggest consolidating these bullet points into a coherent paragraph and emphasizing the key uncertainties this study aims to tackle. To tidy it up, it may help to list the magnitudes of different COS flux components in a table.

**We agree that the bullet points are each very detailed. This is because this part is aimed at the $CO_2$ community, which wants to make use of COS but does not have an extended knowledge of COS sources and sinks. We added Table R2 (Table 1 in the new version of the paper) to consolidate the main information.**

Table R 2: *COS sources and sinks: global estimates available at the time of this study, from Whelan et al., 2018; Remaud et al., 2023.*

| Category | flux component | magnitude ($GgS.y^{-1}$) | references |
|---|---|---|---|
| natural oceanic | source via DMS and $CS_2$ | 265±210 | Lennartz et al., 2017, 2021 |
| | + emissions | 507 | Remaud et al., 2022 (see Sect. 2.2.1) |
| anthropogenic viscose industries | source via $CS_2$ | | |
| anthropogenic others | source via $CS_2$ | 400±180 | Zumkehr et al., 2018 |
| | + emissions | | |
| biomass burning | emissions | 60±37 | Stinecipher et al., 2019 |
| anoxic soils | emissions | 96 | Abadie et al., 2022 |
| oxic soils | net sink | -126 | Abadie et al., 2022 |
| net soils | net sink | -30 | Abadie et al., 2022 |
| | | -89 | Kooijmans et al., 2021a |
| vegetation | uptake | -530 | see Section 2.2.2 |
| | | -664 | Kooijmans et al., 2021a |
| atmospheric oxidation by OH | sink | [-130 , -80] | Whelan et al., 2018 |
| atmospheric photolysis | sink | -50±15 | Whelan et al., 2018 |

L100–L110: This paragraph needs to give a clear roadmap of what the study aims to achieve. Currently, it sounds like this study is picking on the bias in Zumkehr et al. (2018) inventory, but there is more to be said about biogenic COS fluxes and the underlying physiological processes.

**It is true that we cannot access the underlying physiological processes in this study, because we use atmospheric data of mixing ratios of COS, which cannot provide information on these. We only aim at assessing the currently available estimates of COS fluxes. We reformulated to make the text clearer:** *Therefore, this study aims at quantitatively assessing the anthropogenic and biogenic COS fluxes at the regional scale. It demonstrates that a set-up based on one measurement site in Western Europe, which provides data for over half a decade, makes it possible to:*

- *quantitatively assess the discrepancies in Zumkehr's anthropogenic emission inventory in the footprint of the measurement site*

- *evaluate a new inventory based on industrial emission declaration in the European Union*

- *study the seasonal and diurnal cycles of biogenic fluxes, based on the ORCHIDEE and SiB4 processed-based land surface models, and point to strengths and weaknesses in these two models.*

L121–122: How often was the calibration carried out?

**As indicated in Belviso et al. (2020), "Calibration was performed about every three weeks". Information added:** *Calibration is carried out every three weeks using 1-ppm primary standard, [. . . ]*

L134: Here, it would help to refer to Sect. 2.3 for background calculation (L232–238). Or move that paragraph on background calculation here.

**We added a cross-reference to Section 2.3.**

L148: "DMS emissions can only be non natural ..." - Wetlands and the ocean do emit DMS. I get the point of this sentence, but the logic does not follow.

**This sentence was indeed not clear and also contained typos. It has been rewritten as:** *Anthropogenic DMS emissions also exist. However, Sarwar et al. (2023) and von Hobe et al. (2023) have shown that their impact (through oxidation) on simulated COS concentrations is negligible.*

L151: "box models" - I would call them spatially resolved box models to avoid potential confusion.

**This was not clear, we modified the sentence:** *The emissions of the three species have been computed using coarse-resolution box models calibrated [. . . ]*

L165–168: Does this mean that ORCHIDEE and SiB4 COS flux fields used here are optimized posterior estimates?

**ORCHIDEE and SiB4 COS fluxes are not optimized posterior estimates. What was meant here is that the atmospheric COS mixing ratio fields at the surface, which are required by ORCHIDEE and SiB4 to compute the COS fluxes from the surface to the atmosphere, are provided from atmospheric inversions assimilating NOAA COS**

**data, hence representing realistic variability of concentrations. These sentences were confusing and did not bring useful information: they have been removed in the new version of the text.**

L172: "In particular, biogenic fluxes exhibit a significant diurnal cycle." - I am confused here. Aren't the posterior fluxes from Remaud et al. (2022) and Ma et al. (2021) monthly?

**We agree that the sentence is a bit confusing between model and the real world. Actual biogenic fluxes vary diurnally but modelled fluxes do not. We must therefore assess the impact on the simulated concentrations of not taking into account this diurnal cycle. We have clarified the sentence:** *In particular, real-life biogenic fluxes exhibit a significant diurnal cycle (whereas modelled fluxes are constant).*

L175: "We assess the sensitivity of our simulations to daily varying biogenic fluxes by using 3-hourly vegetation uptake as simulated by ORCHIDEE and SiB4 for the year 2016." - Are these posterior flux estimates or unadjusted prior flux estimates?

**These are prior flux estimates, see answer to comment on L165-168 above.**

L183–184: The components sum to 61.4 GgS per year not 62.1 GgS per year. Are there other minor components not listed here?

**Yes, there are: we didn't list the industrial solvents for ≈0.7 GgS nor the paper industry for ≈0.025 GgS, we mention these sub-sectors in the Introduction, in the list of anthropogenic activities, L51seq.**

L186: I think you mean "effective" not "efficient".

**It's been corrected.**

L220: One advantage of a Lagrangian model over an Eulerian model is the flexible resolution at which transport is resolved. As a result, Lagrangian models can use meteorological input at a finer resolution than the resolution at which footprints are aggregated. Is there a reason to run FLEXPART with 1° meteorology input instead of the native resolution (0.25°) of ERA5?

**The main reason to use a coarser meteorology is that the simulations with the native resolution require too much computed resources with regards to the aim of the study, especially as we include a large domain beyond continental Europe to cover oceanic contributions.**

L243: An RMSE of 35 ppt does not feel like a small error; it seems to be around 1/3 of the seasonal cycle amplitude of COS (Fig. 1b).

**We agree, we meant it is small relative to the RMSEs obtained with anthropogenic emissions for example. The sentence has been clarified:** *Its contribution at GIF ensures a good simulation of the variability (Table 2: Pearson's correlation ≥ 0.75) and a base-line mean error ≤ 35 ppt over the whole period of interest.*

L246–248: Again, here it is crucial to clarify whether the three-hourly flux estimates are optimized posterior estimates or prior estimates coming out directly from terrestrial biosphere models.

**See answer to L165-168 above: we hope it is clear now that they are prior estimates from the biosphere models.**

L249: "from 0.74 to 0.72" - The decrease in correlation is a bit surprising because, all else being equal, I would expect the correlation to stay unchanged or increase slightly with the inclusion of diurnal variability. Are these correlation coefficients just for the year 2016 or the entire period?

**The computations with the 3-hourly vegetation in 2016 are done only for the year 2016, we added the same computations for comparison (see Table 1 here) and added the information in the legend of the Table:** *Statistical indicators of the performances of the model compared to the measurements at GIF for each contribution, based on the daily afternoon (12:00-18:00 UTC) means of simulated and measured mixing ratios; indicators are either computed over the whole period, i.e., Aug. 2014 - Dec. 2019 (see Section 2.1) or only over the year 2016 for comparison with the available contribution by the 3-hourly varying vegetation[. . . ]*

L249: "The variability is not better reproduced when adding the natural emissions to the background" - Which component(s) of "natural emissions"?

**The "natural emissions" in this sentence are the ocean and biogenic land, we changed the sentence** *The variability is not better reproduced when adding the natural emissions from the ocean and the biogenic land emissions from the soils and the vegetation to the background (correlations ≤0.75 in all cases), which may be due to a too [. . . ]*

L276–291: Unclear from this paragraph how the contributions of biogenic, anthropogenic, and oceanic COS fluxes to observed COS concentrations compare with each other. I suggest adding a table to list their respective contributions to ΔCOS.

**To help compare the natural and anthropogenic contributions, we added Figure R2 in Appendix B and added in the text:** *At GIF, the seasonal variations of COS mixing ratios are dominated by the contributions of the background and ocean i.e. by large scale fluxes; the variations at shorter time scales (week or day) are driven by the biogenic land contribution (Belviso et al., 2023). Finally, depending on the wind speed and direction, anthropogenic emissions may dominate for short episodes of high concentrations (see section 3.3.2 Selected winter episodes in Belviso et al., 2023). The contributions to COS mixing ratios at GIF due to the ocean, the biogenic land fluxes and the anthropogenic emissions are shown in Figure B1.*

[Figure]

**Figure R 2.** *Occurrences of contributions ($\Delta$COS, in ppt) to COS mixing ratios at GIF due to the background, the ocean, the vegetation and soil ("biogenic land") and our home-made anthropogenic emissions (total and only the Paris+Rouen area). Total number of occurrences: 1673 simulated values matching afternoon (12:00-18:00UTC) valid mean observations between 2014 and 2019.*

Fig. 1: In panel a, it's quite challenging to distinguish between GIF (afternoon) from GIF (all). Consider enhancing the color contrast or use different symbols. In the caption, background calculation should refer to Sect. 2.3 not 2.1.

**We enhanced the color contrast and increased the size of the figure; the cross-reference has been corrected.**

215   Fig. 2: In panel e, it would be more helpful to show the summed footprint map over the entire study period to check the influence area coverage. Readers may not know if the footprint map for May 2019 is representative of the whole period (2014–2019).

**The footprint for May 2019 shown in the Figure is an illustration of the typical size of the area covered by the footprint. We put the footprints summed up over a whole year in the new Figure 2.**

220   Fig. 4: Unclear what "growths" means in the figure legend. For panel c, it would be better to report COS fluxes in pmol m$^{-2}$ s$^{-1}$.

**Figure 4 has been changed accordingly.**

**2 Report #2 by Anonymous referee #4**

**3 General comments**

225 This is an interesting paper that performs a detailed analysis of a long time series of COS measurements in GIF, France. Although the wording should be improved (I had difficulties understanding the manuscript at several places), I particularly like the systematic analysis of the impact of different fluxes on the simulated mixing ratios. This allows the authors to draw conclusions about the quality of the anthropogenic emission inventory, and the performance of the biosphere model. Nevertheless, I would like to suggest a couple of checks and/improvements that could further improve the understanding.

230 (1) CS2 is emitted by industry (VI in their homemade inventory). CS2 is converted in the model to COS with a prescribed timescale. It would be instructive to test (a) the impact of the implied timescale (conversion rate is not very well known and depends on season) (b) the impact of a diurnal variation on the CS2 $\rightarrow$ COS conversion. Since the conversion is performed by the OH radical, these impacts are vital to analyze

**We agree that the lifetime of $CS_2$ is not well known. We tested two of the available values for the kinetics of the**
235 **oxidation of $CS_2$ by OH: the conclusions of our study did not change. We added this information in Section 2.3:** *Note that a life time of 1.5 day has also been tested (not shown), with almost no changes in the conclusions of this study.*

**We agree that a study targeting the diurnal variations would need to take into account the impact of OH diurnal cycle. Since diurnal varying emissions were not available at the time of our study, save for 3-hourly vegetation fluxes for 2016, the impact of chemistry at the hourly resolution is not in the scope of our study. We added this item in the**
240 **Conclusion:** *For anthropogenic and oceanic $CS_2$ emissions, the kinetics of the conversion of $CS_2$ into COS through its ts oxidation by OH is not well-known. Moreover, through OH availability and temperature, the lifetime of $CS_2$ depends on the season and has a diurnal cycle. The impacts of these variations on the chemical source of COS must be assessed, when more relevant information, e.g. varying diurnal emissions are available.*

(2) The model uncertainty is not discussed in any detail. The authors employ a Lagrangian particle model, and it would be
245 instructive to know what is the impact of the number of particles released every six hours, the effect of resolution on the results, etc. Since the analysis is based on only one station, a proper representation of that station in the model is important. How well is the planetary boundary layer height simulated? What about wind validation?

**The Lagrangian model used in this study, FLEXPART, plus the meteorological data on which it relies, ECMWF ERA5, are well recognized tools in the atmospheric community. Moreover, the boundary layer height and wind direc-**
250 **tions are provided by the state-of-the-art ECMWF ERA5 data with very good performances (e.g., Molina et al., 2022). We added more information regarding our choices of configuration in Section 2.3:** *The performances of ERA5-provided boundary layer height and wind direction are very good (e.g. Molina et al., 2021).* **and** *FLEXPART and ERA5 meteorological data are well recognized tools in the atmospheric community, the combination of the two being widely used in the atmospheric inversion community (Pisso et al., 2019; Bakels et al., 2024). The choices made for FLEXPART's configuration (number of*
255 *particles released, frequency of releases, resolution) are consistent with the set ups used for the atmospheric inversions fluxes (e.g., Bergamaschi et al., 2022; Thompson et al., 2021).*

(3) The final analysis of the paper investigates concentration changes between 6 a.m and 6 p.m., and between 6 p.m. and 6 a.m. the next day. In spring the diurnal variation in the biogenic fluxes greatly improves the comparison to simulations. In summer, this is not the case, but this is not analyzed further. My first guess would be that the effects on the diurnal fluxes interfere with the (changing) effect of entrainment from the atmosphere aloft. I would suggest to perform a budget analysis in which the concentration changes are attributed to (1) the effect of the fluxes (2) the effects of vertical transport (i.e. entrainment or detrainment) and (3) advection.

**We agree that such a study would be interesting but it would also be very complex and not in the scope of this paper. It is further work required to fully make use of the long COS mixing ratios time series. We added this item as a perpective:** *An analysis discriminating between the impacts of i) the fluxes, ii) the chemical source, iii) the vertical transport and iv) the advection on COS mixing ratios could be performed from a set of simulations, with a careful design, computer resources and up-to-date hourly varying fluxes.*

**4    Specific comments**

Further remarks and suggestions are provided in the annotated pdf.

**These remarks are detailed here with our answers.**

- 10-13: Too long sentence

   **OK** *The main limitation of this inventory is the flat temporal variability applied to anthropogenic fluxes due to lack of information on industrial and power-generation activity in viscose factories and in coal-power plants. As a consequence, there are potential mismatches in the simulated plumes emitted from these hot-spots.*

- 27: retrieve: estimate (is better I think)

   **OK**

- 27,28: in the prior and in the measured: by (or cintained in)

   **OK**

- 61: 96 GgS: here no uncertainty?

   **There is indeed no uncertainty provided by the process-models.**

- 94-96->issue: unclear sentence

   **We clarified the sentence:** *Anthropogenic emissions in the footprint of GIF were proven to be overestimated when analyzing specific events selected in the relatively long continuous time series available. Nevertheless, further characterization was not possible at the time of the study.*

- 112: purposes (or which is the main original purpose)

   **OK**

– 122: At what mixing ratio level? 1.2% of 1 ppm is not OK!

**Instrument and calibration information are extensively detailed in Belviso et al., 2013 and 2016. Information on calibration was misleading in the previous version of the manuscript. We update it as follows:** *Calibration and drift correction is done every three weeks using a calibration gas containing 1.013 ppm of COS in helium, with occasional calibration using a standard of compressed air with 573 ppt of COS, and another standard traceable to NOAA ESRL standard of 448.6 ppt. Calibrations led to a repeatability of 1 % and a precision of 0.2 %.*

– 136: designed -> need specification: was this a global inversion?

**Yes, it was, it has been made clearer:** *These global atmospheric inversions were designed to assimilate data from the background NOAA observation sites, such as MHD.*

– 148 DMS emissions: ??? You mean emissions from land? As far as I know ocean emissions are natural.

**This sentence was not clear at all and contained typos. It has been rewritten as:** *Anthropogenic DMS emissions also exist. However, Sarwar et al. (2023) and von Hobe et al. (2023) have shown that their impact (through oxidation) on simulated COS concentrations is negligible.*

– 166-168 -> fluxes: Unclear: you use fluxes to compute fluxes? I think you use mixing ratios to compute first order uptake (and not the fixed 500 ppt "placeholder". **What was meant here is that the atmospheric COS mixing ratio fields at the surface, which are required by ORCHIDEE and SiB4 to compute the COS fluxes from the surface to the atmosphere, are provided from atmospheric inversions assimilating NOAA COS data. These sentences were confusing and did not bring useful information: they have been removed in the new version of the text.**

– 176: is: should be are, but maybe assume "performance" instead of performances.

**OK**

– 187 no necessarily -> NOT

**OK**

– 200: viscose: But how do you account for the indirect nature of CS2 emissions? Oxidation to COS is not instantaneous....

**A cross-reference to the description of how our simulations account for CS$_2$ oxidization, in Sect.2.3, has been added.**

– 213 Beauvais: Why is this relevant? Is there industry there?

**We have made the text clearer:** *The closer to GIF (large black triangle in Northern France in Figure 2c) is in the city of Beauvais, which lies about 85 km to the North of GIF, i.e., further than the Paris area.*

– 224: 3 days: OK, maybe mention this earlier in the manuscript somewhere....One more issue on this: CS2 is oxized y OH which has a distinct diurnal cycle: does that matter?

**See answer to L200. For the diurnal cycle, please see the answer to general comment 1.**

- 251: what is "coarse variability"?

  **Here we mean that the monthly temporal resolution of the oceanic emissions is too low compared to the variability of the transport. This sentence has been modified as:** *The variability is not better reproduced when adding the natural emissions from the ocean and the biogenic land emissions from the soils and the vegetation to the background (correlations ≤0.75 in all cases). This may be due to the monthly resolution of the ocean emissions (Section 2.2.1) being too coarse compared to the variability of the transport and to an issue with the variability of the vegetation uptake or the soil exchanges, probably at the seasonal scale, which is assessed in Section 3.3.*

- 270 further invest are -> IS

  **OK**

- 292 obs sites: here it would also help to analyse stack emissions directly?

  **We agree that emission factors used to assess coal-power-plant emissions are very coarse and can vary a lot depending on the power-plant technology, the type, origin and quality of coal used in the plant and other factors. Similarly, we used CS2 emissions as declared and reported by industries, also based on imprecise emission factors. Having extra measurements directly targeted at specific industry would significantly improve our understanding of anthropogenic emissions. In Belviso et al., 2023, we used measurements carried out directly downstream a viscose factory in Rouen to evaluate emissions from this factory. More systematic studies would be needed, although mobile campaigns are complex to operate considering the complexity of COS measurements. We include such short discussion in the text**. *Additional continuous observation sites would be needed in different places of Europe to clarify the relevancy of our inventory beyond the Paris area and vicinity. In addition, our inventory is based on coarse emission factors, both for the viscose industry and for coal power plant. Facility-level campaign as carried out around a viscose factory near Rouen in Belviso et al., 2023, would significantly improve our understanding of COS anthropogenic fluxes, thus our inventory.*

- 301: between day D at 6 p.m. and day D at 6 a.m.: strange that you mention the later time first? Maybe swap?; variation: I think "variation" should be better defined. What does it exactly mean?; I guess I get it now (later), but I think "a mean concentration difference between xx and xxx would be much clearer. Variation for me is related to variability. + 307: I must say that variation of variations does not clarifies the situation here? Please define a metric, and give it a name (DV, NV)??? but define properly

  **We changed "variation(s)" to "difference(s)" and introduced the notations $\Delta_{day}^{\text{COS}}$ and $\Delta_{night}^{\text{COS}}$ where relevant in Section 3.3 and Figure 4.**

- 305: have impact on, or impact (without on x2).

  **OK**

- 322: consistently->consistent

  **OK**

– 327: uptake: this last addition is unclear....I guess you want to say that plant uptake during night is the cause?

**This sentence was not clear and has been changed to:** *This discrepancy suggests a persistent nighttime uptake not adequately represented by ORCHIDEE's diurnal cycle. The ORCHIDEE minimal stomatal conductance at night (see Section 2.2.2) has to be revised.*

– 332: Sib4: . Here, a budget analysis would help. How much of the increments are caused by land-atmosphere exchange, and how much by transport processes mixing the boundary layer (or stabilizing it)? **Such an analysis is beyond the scope of this first study: please see answer to general comment 3.**

– 346, 348, 350: declared -> reported

**OK**

– 351: arise -> arises? or limitations arise

**OK**

– 360: meteorological: would a "plume" approach help here? Aermod type?

**This study does not treat with scales for which plume based models are relevant. Independently of the quality of the model, a simple shift in the wind direction or speed in the meteorological data leads to high discrepancies in the mixing ratios. Therefore a plume approach would not be less dependent on the transport limitations.**

– 367: winter at night: here the question is whether the model is adequately able to represent the stable night-time boundary layers in winter...

**The nighttime BL in winter is probably not very well represented, even though we use state-of-the-art ECMWF data. These uncertainties prevent us from quantifying the under-estimation of the nighttime COS uptake; however, our night-day-based approach confirms qualitatively the existence of this under-estimation. We clarified this in the text** *The uncertainties in the modeling of transport at night in winter prevent us from quantifying the underestimation of the nighttime COS uptake. However, the discrepancies between simulations and observations shown by the nighttime and day differences suggest the existence of such an under-estimation.*